

# Information Content of Brightness Temperature Differences of Spaceborne Imagers with respect to Cloud Phase

Johanna Mayer[a], Bernhard Mayer[b,a], Luca Bugliaro[a], Ralf Meerkötter[a], and Christiane Voigt[a,c]

[a]Deutsches Zentrum für Luft- und Raumfahrt, Institut für Physik der Atmosphäre, Oberpfaffenhofen, Germany
[b]Ludwig Maximilians Universität, Institut für Meteorologie, Munich, Germany
[c]Johannes Gutenberg-Universität, Mainz, Germany

**Correspondence:** Johanna Mayer (johanna.mayer@dlr.de)

**Abstract.** This study investigates the sensitivity of two brightness temperature differences (BTDs) in the infrared (IR) window of the SEVIRI imager to various cloud parameters in order to better understand their information content, with a focus on cloud thermodynamic phase. To this end, this study presents radiative transfer calculations, providing an overview of the relative importance of all radiatively relevant cloud parameters, including thermodynamic phase, cloud top temperature (CTT), optical thickness ($\tau$), effective radius ($R_{eff}$) and ice crystal habit. By disentangling the roles of cloud absorption and scattering, we are able to explain the relationships of the BTDs to the cloud parameters on the one hand by spectral differences in the cloud optical properties. In addition, an effect due to the nonlinear transformation from radiances to brightness temperatures contributes to the specific characteristics of the BTDs and their dependence on $\tau$ and CTT. We find that the dependence of the BTDs on phase is more complex than sometimes assumed. Although both BTDs are directly sensitive to phase, this sensitivity is comparatively small in contrast to other cloud parameters. Instead, the primary link between phase and the BTDs lies in their sensitivity to CTT, which is associated with phase. One consequence is that distinguishing high ice clouds from low liquid clouds is straightforward, but distinguishing mid-level ice clouds from mid-level liquid clouds is challenging. These findings help to better understand and improve the working principles of phase retrieval algorithms.

## 1 Introduction

Passive spaceborne imagers, with their wide field of view and, in the case of geostationary satellites, high temporal resolution, allow global observations of clouds. These passive instruments typically use solar and/or infrared (IR) window channels to retrieve cloud properties. The advantage of pure IR-based retrievals is that they can be applied during both daytime and nighttime (Nasiri and Kahn, 2008; Cho et al., 2009). Such IR retrievals often use brightness temperature differences (BTDs) of IR window channels, for instance to detect clouds or retrieve cloud properties like optical thickness ($\tau$) or effective particle radius ($R_{eff}$) (e.g., Inoue, 1985; Krebs et al., 2007; Heidinger et al., 2010; Garnier et al., 2012; Kox et al., 2014; Vázquez-Navarro et al., 2015; Strandgren et al., 2017).

Another cloud parameter which is often retrieved using BTDs is the cloud thermodynamic phase (ice, liquid, mixed) (Ackerman et al., 1990; Strabala et al., 1994; Finkensieper et al., 2016; Key and Intrieri, 2000; Baum et al., 2000, 2012; Mayer et al., 2024). Accurate satellite retrievals of cloud phase are important for various reasons. Firstly, the cloud phase plays an important





role in cloud-radiation interactions (Komurcu et al., 2014; Choi et al., 2014; Matus and L'Ecuyer, 2017; Ruiz-Donoso et al., 2020; Forster et al., 2021; Cesana et al., 2022). Several studies highlight its impact on climate sensitivity within general circulation models (Gregory and Morris, 1996; Doutriaux-Boucher and Quaas, 2004; Cesana et al., 2012; Tan et al., 2016; Bock et al., 2020). Accurate observations of the cloud phase are thus essential to improve cloud representation in climate models (Cesana et al., 2015; Atkinson et al., 2013; Matus and L'Ecuyer, 2017; Bock et al., 2020). Additionally, determining cloud

phase is a necessary step in the remote sensing retrieval of cloud properties, including $\tau$, $R_{eff}$ and water path (Marchant et al., 2016).

However, determining cloud parameters such as the thermodynamic phase from BTDs is a challenging task. Radiative transfer through clouds and the atmosphere is complex, with many parameters that can in principle influence satellite observations. The relative importance of these parameters is often not fully understood.

Ackerman et al. (1990) were the first to observe a correlation between BTDs in High-Resolution Interferometer Sounder (HIS) data and the different cloud phases as determined by concurrent lidar data. They proposed a trispectral technique to distinguish between ice, water, and clear sky using the BTDs between channels at about 8 $\mu$m and 11 $\mu$m (BTD(8.0-11.0)) and between channels at about 11 $\mu$m and 12 $\mu$m (BTD(11.0-12.0)). Strabala et al. (1994) expanded on their findings, using MODIS airborne simulator data. They considered clouds of varying $\tau$ and found that distinguishing between ice and water clouds using

these BTDs is difficult for optically thin clouds. Parol et al. (1991) and Dubuisson et al. (2008) studied the sensitivity of BTDs to effective radius $R_{eff}$ and particle shape for cirrus clouds. Parol et al. (1991) found that the BTD(11.0-12.0) for the Advanced Very High Resolution Radiometer AVHRR aboard the NOAA satellites is sensitive to whether cloud particles are spherical or non-spherical. Dubuisson et al. (2008) showed that the impact of different non-spherical ice crystal shapes on BTD(10.6-12.0) and BTD(8.7-10.6) of the Infrared Imaging Radiometer IIR aboard CALIPSO is small compared to their sensitivity to $R_{eff}$.

The effect of $R_{eff}$ on the BTDs was also considered by Baum et al. (2000), who further extended the trispectral method for MODIS phase retrievals by incorporating information about the horizontal variability of the BTDs. Similar to the study of Strabala et al. (1994), the radiative transfer simulations of Baum et al. (2000) primarily focused on low-level water clouds and high cirrus clouds, and did not consider midlevel clouds. To bridge this gap, Nasiri and Kahn (2008) conducted a sensitivity study that considered also midlevel clouds for the MODIS BTD(8.5-11.0). They showed that BTD(8.5-11.0) is sensitive to

cloud top height (CTH) and that this leads to limitations in the phase discrimination in the cloud temperature regime where both liquid and ice can exist.

These studies show that many different parameters influence the BTDs: Cloud parameters considered in previous studies include thermodynamic phase, $\tau$, $R_{eff}$, ice crystal habit, and CTH. As outlined above, most of the studies so far have however each focused on only a small number of these cloud parameters; an overview over the relative importance of all these cloud

parameters is still missing. Especially the influence of CTH or cloud top temperature (CTT) on BTDs has not been studied in detail, with exception of Nasiri and Kahn (2008). Besides cloud parameters also the amount of water vapour in the atmosphere (mainly above the clouds) affects BTDs even in the (relatively) transparent spectral window region 8–12 $\mu$m. This has been pointed out by several authors (Strabala et al., 1994; Nasiri and Kahn, 2008; Dubuisson et al., 2008), but the relative importance of atmospheric absorption compared to cloud parameters on BTDs has not been studied systematically.




In addition, the origin of the dependence of BTDs on cloud thermodynamic phase, as observed in satellite measurements and radiative transfer results, is not fully understood. For phase retrievals it is usually argued that variations in the refractive indices of ice and water across the infrared window cause the BTDs to be sensitive to cloud phase (Finkensieper et al., 2016; Key and Intrieri, 2000; Baum et al., 2000, 2012). However, besides these effects of the cloud phase, the phase also correlates with other cloud parameters like CTT and $R_{eff}$, which in turn have large effects on the BTDs as mentioned above. It is not

fully understood which cloud parameters dominate the response of the BTDs in given cloud scenarios. Additionally, traditional explanations of the phase dependence of BTDs have neglected scattering effects, which as we will show can be substantial. Thus, it is not well understood which physical processes are responsible for the observed phase dependence of the BTDs. A full understanding of the satellite channel dependencies is however critical to design optimal cloud (phase) retrievals and to understand their limitations.

To compute BTDs, satellite radiances are first transformed into brightness temperatures (BT). This transformation by means of Planck's radiation law is a nonlinear function. As nonlinear functions can lead to unexpected behaviour, we expect that there are some effects of the nonlinear relationship between satellite radiances and BTs on BTDs. To our knowledge, the effect of this nonlinear relationship has not been analysed before.

We use Radiative Transfer (RT) calculations to study two BTDs of the SEVIRI imager aboard Meteosat Second Generation

(Schmetz et al., 2002): The BTDs between the IR window channels centered at 8.7 and 10.8 $\mu$m (BTD(8.7-10.8)) and between those centered at 10.8 and 12.0 $\mu$m (BTD(10.8-12.0)). These are the BTDs that are mainly used to identify cloud top phase and determine (ice) cloud properties. First, we investigate the effect of the nonlinear relationship between radiances and BTs on the BTDs. We then use the RT calculations to analyse dependencies and sensitivities of the BTDs with respect to all radiatively important cloud parameters, namely phase, CTT, $R_{eff}$, ice crystal habit and optical thickness ($\tau$) at 550 nm, disentangling

effects of cloud particle scattering and absorption. We also consider the effect of water vapor in the atmosphere on BTDs by comparing the computed BTDs with scenarios without molecular absorption. The findings of these RT calculations are then used to analyse the information content of the BTDs with respect to cloud phase. Overall in this study we focus on the effect of cloud parameters; the effects of other parameters like viewing angle, surface emissivity or atmospheric temperature profiles are not studied.

The aim of this study is twofold: First, it provides an analysis of the effects of all cloud parameters on the two BTDs, disentangling the interactions among the different parameters. Second, this study improves the physical understanding of the role of the different radiative processes leading to different BTD values. This helps to understand the information content of the BTDs with respect to the thermodynamic phase in order to better understand and improve the working principles of phase retrieval algorithms and to understand their uncertainties and limitations. We focus on the phase, but our results are also useful

to better understand the dependencies of BTDs for other remote sensing applications where they are typically used, such as the retrieval of $\tau$ and $R_{eff}$. Since BTDs also depend on atmospheric and surface parameters whose effects are not studied here, this study does not aim at explaining every phenomenon encountered with BTDs. However, understanding the effects of the cloud parameters helps to disentangle different physical cloud-related processes in all atmospheric or surface conditions.





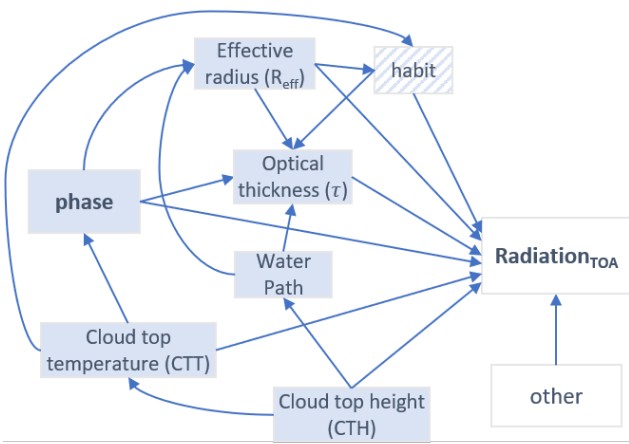

**Figure 1.** Causal diagram of cloud parameters that are connected to the cloud phase. Arrows indicate causal links.

## 2 Physical background

To visualize relationships and dependencies between radiation at top of atmosphere (TOA) and cloud properties, the representation in form of a causal diagram is very useful. Figure 1 shows cloud parameters that are related to the cloud phase, connected by arrows indicating causal relationships. Other factors influencing the radiation at TOA (in particular passive satellite observations), like solar and satellite viewing angles, surface temperature or atmospheric properties, are summarized under "other" in the diagram.

In this paper we use the terms "direct" and "indirect" influence of the cloud phase on the TOA radiation. Direct influence means the effect of changing the cloud phase while all other cloud parameters ($R_{eff}$, CTT, $\tau$, ...) remain the same (represented by the arrow from phase to TOA radiation in Fig. 1). The indirect influence of the cloud phase is represented by all other paths from phase to TOA radiation in Fig. 1. For example, the phase affects $\tau$ and $R_{eff}$, which in turn affect TOA radiation. Information on these two parameters can give an indication about the cloud phase – e.g. clouds with small $R_{eff}$ are typically liquid clouds; clouds with very low $\tau$ are typically ice clouds. Ice crystal habit can influence the TOA radiation as well, but is of course only relevant for ice clouds. The ice crystal shape depends on $R_{eff}$, since certain shapes are more common for smaller/larger $R_{eff}$. CTT and CTH are closely related variables that influence radiation through temperature-dependent cloud emissions and by affecting the atmospheric column above the cloud that can absorb radiation, respectively. CTT is critical for phase determination since for temperatures above $0°$C only liquid and below $-40°$C only ice is physically possible. Between these thresholds, the probability for ice (liquid) clouds increases (decreases) as CTTs get colder (Mayer et al., 2023).

In order to calculate the radiative transfer through a cloud with given cloud (microphysical) parameters, it is necessary to know how much radiation is absorbed, scattered and emitted, i.e. the optical properties of the cloud. The translation from cloud (microphysical) parameters to optical parameters is given by the so-called *single scattering properties*. Mathematical expressions for the TOA radiation as a function of the single scattering properties can be found in appendix A. The single scattering





properties are the volume extinction coefficient $\beta_{\text{ext}}$, the single scattering albedo $\omega_0$ and the scattering phase function $p$. The volume extinction coefficient $\beta_{\text{ext}}$ describes how much radiation is removed through scattering and absorption (=extinction) from a ray when passing through the cloud and can be expressed as

$$\beta_{\text{ext}} = \beta_{\text{sca}} + \beta_{\text{abs}} \tag{1}$$

where $\beta_{\text{sca}}$ and $\beta_{\text{abs}}$ are the scattering and absorption coefficient, with units of m$^{-1}$, measuring how much radiation is absorbed and scattered by cloud particles. Note that in this study $\tau$ is $\beta_{\text{ext}}$ at wavelength $\lambda = 550\,\text{nm}$ integrated over the path through the cloud; the optical thickness $\tau_\lambda$ at other wavelengths $\lambda$ is in general different from $\tau$, depending on the other microphysical cloud parameters. The single scattering albedo $\omega_0$ is a measure of the relative importance of scattering and absorption, defined as

$$\omega_0 = \frac{\beta_{\text{sca}}}{\beta_{\text{sca}} + \beta_{\text{abs}}} = \frac{\beta_{\text{sca}}}{\beta_{\text{ext}}}. \tag{2}$$

Hence, as an alternative to $\beta_{\text{ext}}$ and $\omega_0$ one can equivalently describe radiative transfer by $\beta_{\text{abs}}$ and $\beta_{\text{sca}}$, which can be easier to interpret. The scattering phase function $p(\Omega)$ gives the probability of the scattering angle $\Omega$, i.e. the angle between the incident radiation and the scattered radiation. To understand radiative transfer through a cloud, the most important property of $p$ is the angular anisotropy of the scattering process. This anisotropy is indicated to first order by the asymmetry parameter $g$, which is calculated from $p$ as the mean cosine of the scattering angle $\Omega$.

$$g = \int\limits_{-1}^{1} p(\cos\Omega')\cos\Omega'\,d\cos\Omega' \tag{3}$$

If a particle scatters more in the forward direction ($\Omega = 0°$), $g$ is positive; $g$ is negative if the scattering is more in the backward direction ($\Omega = 180°$) (Bohren and Huffman, 2008). The interplay of the single scattering properties $\beta_{\text{ext}}, \omega_0$ and $p$, in combination with the cloud water path, determines how much radiation is transmitted through a cloud and, in combination with the cloud temperature, how much radiation is emitted from it. The single scattering properties depend on the wavelength of the radiation and on the cloud parameters R$_{\text{eff}}$, habit and phase. They are shown in Fig. 2 for varying R$_{\text{eff}}$ and cloud phase. The variations of the single scattering properties due to habit are mostly small in comparison and therefore not shown. Instead of $p$ we show $g$ as a simpler measure to characterize the scattering process. The single scattering properties for ice are computed according to Baum et al. (2011), for liquid droplets according to Mie theory.

The spectral variations of $\beta_{\text{abs}}$, $\beta_{\text{sca}}$ and $g$ translate into different BTD values for different cloud parameters. This will be investigated in detail in the next sections using radiative transfer calculations, but we can already derive some predictions for the BTDs by comparing $\beta_{\text{abs}}$, $\beta_{\text{sca}}$ and $g$ at the three IR wavelengths (shown in Fig. 2) when $\tau$ is small enough such that transmission through the cloud dominates over cloud emission. First, we take a look at the effects of absorption. Since $\beta_{\text{abs}}$ increases over the three channel wavelengths for both phases and all R$_{\text{eff}}$, absorption is always stronger at 12.0 than at 10.8 and stronger at 10.8 than at 8.7 $\mu$m. This means that we expect positive values for both BTDs due to absorption, when transmission through the cloud is the dominant process compared to emission. The spectral variation of $\beta_{\text{abs}}$ is larger for smaller R$_{\text{eff}}$. Hence,





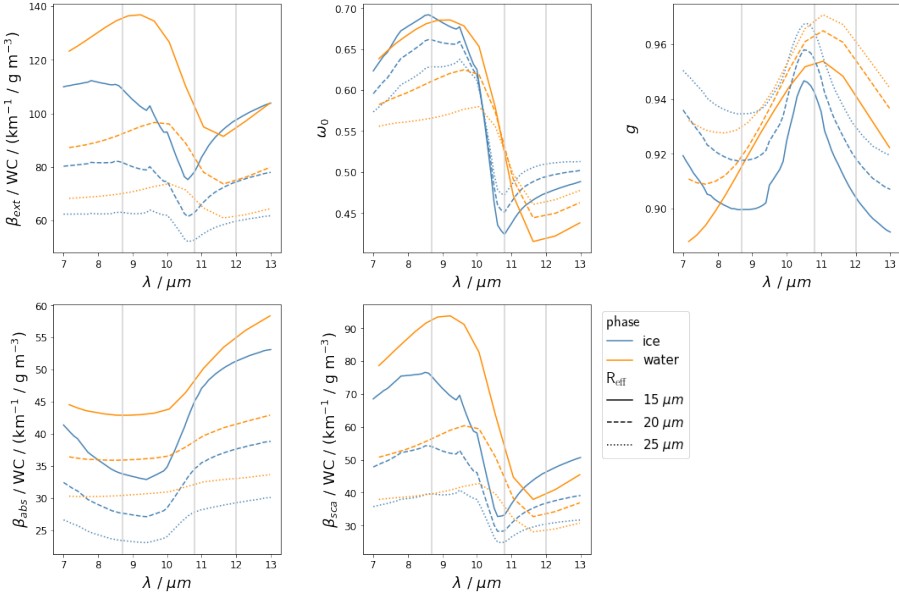

**Figure 2.** Single scattering properties extinction coefficient $\beta_{\text{ext}}$, single scattering albedo $\omega_0$ and asymmetry parameter $g$, as well as absorption coefficient $\beta_{\text{abs}}$ and scattering coefficient $\beta_{\text{sca}}$ (computed from $\beta_{\text{ext}}$ and $\omega_0$) as functions of wavelength for varying cloud phase and effective radius $R_{\text{eff}}$. $\beta_{\text{ext}}$, $\beta_{\text{abs}}$ and $\beta_{\text{sca}}$ are scaled by the cloud water content WC. Parameterisations for ice according to Baum et al. (2011), for liquid droplets according to Mie theory. For ice clouds, the "general habit mix" was used as ice crystal habit. Vertical grey lines indicate the centre wavelengths of the three IR window channels.

for two clouds with the same water content but different $R_{\text{eff}}$ we expect larger BTD values for the cloud with the smaller $R_{\text{eff}}$. Comparing the two cloud phases, the spectral variation in $\beta_{\text{abs}}$ between 8.7 and 10.8 $\mu$m is stronger for ice than water particles, meaning that we expect larger BTD(8.7-10.8) values for ice than for water clouds due to absorption - given that $R_{\text{eff}}$ is the same for the ice and water cloud. Between 10.8 and 12.0 $\mu$m the spectral variation in $\beta_{\text{abs}}$ is similar for ice and water particles. It is more challenging to derive predictions about scattering, since this is governed not by a single quantity but by the interplay of $\beta_{\text{sca}}$ and $g$. The spectral variation of $\beta_{\text{sca}}$ implies that scattering is stronger at 8.7 than at 10.8 or 12.0 $\mu$m. However, scattering has a stronger effect on the transmission of radiation when it is not only concentrated in the forward direction ($g$ values close to 1) but also in other directions (smaller values of $g$). Where $g$ has smaller values for ice than water particles (see Fig. 2), scattering has a larger effect on the transmission of radiation for ice than for liquid clouds (see also Parol et al. (1991)).

## 3 Radiative transfer calculations

Simulations for the three IR window channels of the SEVIRI instrument centered at 8.7, 10.8 and 12.0 $\mu$m were performed for a variety of water and ice clouds using the sophisticated radiative transfer package libRadtran (Mayer and Kylling, 2005; Emde et al., 2016; Gasteiger et al., 2014). LibRadtran represents water and ice clouds in detail and realistically. It has been





validated against observations and in several model intercomparison campaigns and has been extensively used to develop or
validate remote sensing retrievals (e.g. Mayer et al., 1997; Meerkötter and Bugliaro, 2009; Bugliaro et al., 2011; Stap et al.,
2016; Piontek et al., 2021b; Bugliaro et al., 2022). The optical properties of water droplets are calculated using Mie theory.
For ice crystals, we use the Baum et al. (2011) parameterization of optical properties for three different habits (general habit
mixture, columns, rough aggregates). Simulations of TOA radiances for the SEVIRI IR window channels are made using the
one-dimensional radiative transfer solver DISORT (Discrete Ordinate Radiative Transfer) 2.0 by Stamnes et al. (2000); Buras
et al. (2011). The complete permutation of $\tau$, $R_{eff}$, CTT/CTH, crystal habits and phase was simulated and is listed in table 1.
The CTT is set to the atmospheric temperature at the altitude of the CTH and represents the temperature at cloud top. For
simplicity we keep the cloud geometric thickness constant at 1 km; the impact of variable geometric thickness is discussed in
Sect. 6.3. We only consider single-phase (ice or water) and single layered clouds. True mixed-phase clouds and multilayered
clouds are not considered.

The simulation setup in terms of atmosphere, satellite/solar geometry and surface type is summarized as well in table 1.
As we focus on the influence of cloud parameters in this study, the atmospheric parameters, surface parameters and satellite
geometry are kept constant for all simulations. We use the US standard atmosphere (Anderson et al., 1986) and a surface
temperature of 290 K. We place the simulations over the ocean where the surface emissivity is nearly constant for the three
IR window channels and set it to 1. The satellite zenith angle (SATZ) is kept constant at 0° (nadir view). Implications of
generalising our results to other atmospheres and satellite setups are discussed in Sect. 6.3.

To disentangle cloud effects from effects of the atmosphere, we also compute simulations with molecular absorption
switched off. LibRadtran further has the possibility to simulate the IR window channels for cloud layers for which scattering is switched off, meaning that the scattering coefficient in the simulation is set to zero while the absorption coefficient
remains constant. This allows to disentangle effects of scattering and absorption in a cloud.

## 4 Effects of Planck's law: the BTD Nonlinearity Shift

Before analysing the results of the RT calculations, we examine the effects of the nonlinear relationship between radiances and
BTs on the BTDs. We call these effects *BTD Nonlinearity Shift*. The BTD Nonlinearity Shift is purely due to the nonlinearity
in the computation of BTDs and not due to wavelength dependent optical properties of the cloud, which we will focus on in
the next sections of this study. BTDs are calculated from measured radiances using Planck's radiation law, which describes the
spectral radiance $B_\lambda$ of a black body emitting radiation at temperature $T$

$$B_\lambda(T) = \frac{2hc^2}{\lambda^5}(e^{\frac{hc}{\lambda k_B T}} - 1)^{-1},\tag{4}$$

where $h$ is the Planck constant, $c$ is the speed of light in vacuum and $k_B$ is the Boltzmann constant. The inverse Planck function
accordingly maps spectral radiance $R_\lambda$ to the corresponding temperature

$$T_\lambda(R_\lambda) = \frac{hc}{k_B \lambda} \frac{1}{\ln(\frac{2hc^2}{\lambda^5 R_\lambda} + 1)}\tag{5}$$

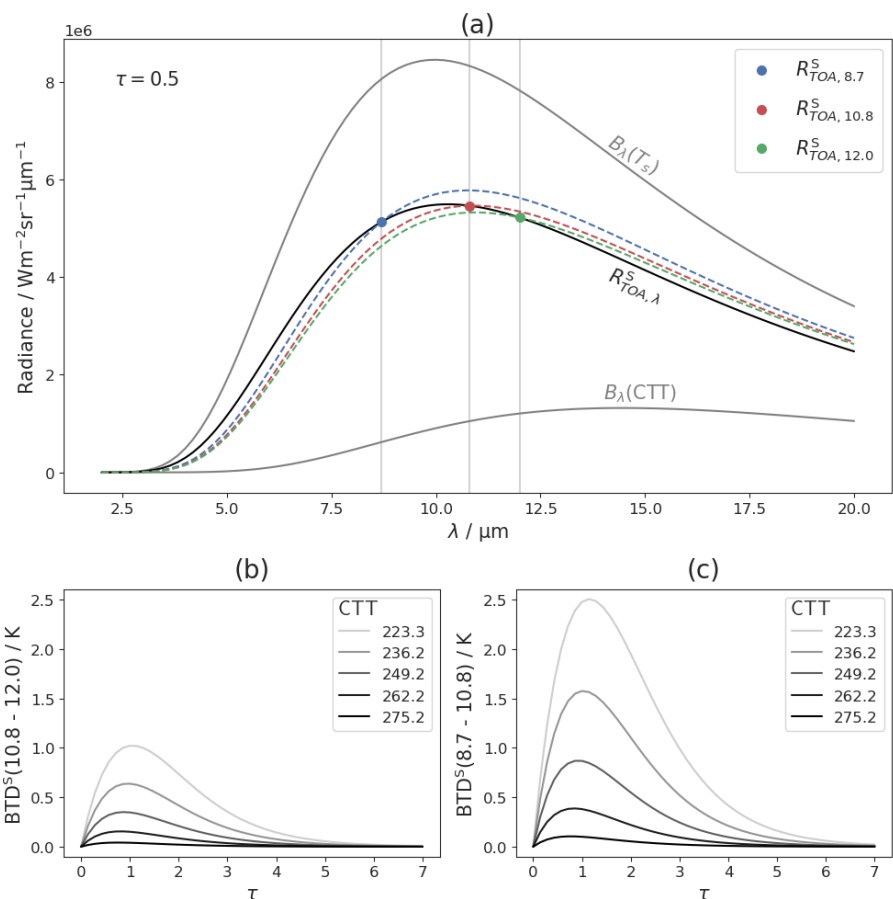

**Figure 3.** (a) Radiance at top of atmosphere ($R_{\mathrm{TOA},\lambda}^{\mathrm{S}}$) computed with the Schwarzschild equation (black line). Vertical grey lines indicate the centre wavelengths of the three IR window channels with blue, red and green dots at $R_{\mathrm{TOA},8.7}^{\mathrm{S}}$, $R_{\mathrm{TOA},10.8}^{\mathrm{S}}$ and $R_{\mathrm{TOA},12.0}^{\mathrm{S}}$ respectively. The blue, red and green dashed lines correspond to the Planck curves of these three TOA radiances, i.e. $B_\lambda(T_\lambda(R_{\mathrm{TOA},\lambda}^{\mathrm{S}}))$ for each wavelength, where $B_\lambda$ is the Planck function and $T_\lambda$ the inverse Planck function. The grey solid curves show the Planck curves of the surface temperature $T_s$ and the CTT as reference. (b) Brightness temperature differences computed with the Schwarzschild equation, $\mathrm{BTD}^{\mathrm{S}}$, as functions of $\tau$ for different CTTs and a fixed $T_s$=290 K.





**Table 1.** Setup and cloud Properties for libRadtran radiative transfer calculations (SATZ = satellite zenith angle, SKT = skin temperature)

| cloud properties | |
|---|---|
| phase | liquid, ice |
| $R_{\text{eff}}$ (liquid clouds) | 5, 10, 15, 20 $\mu$m |
| $R_{\text{eff}}$ (ice clouds) | 20, 30, 40, 50 $\mu$m |
| $\tau$ | 0, 0.1, 1, 2, 3, 5, 7, 10, 15, 30 |
| ice habit | general habit mix (ghm), rough aggregates, solid columns |
| optical properties | for ice after Baum et al. (2011) for liquid droplets Mie |
| CTH (liquid clouds) | 1, 2, 4, 6, 8 km |
| CTT* (liquid clouds) | 281.7, 275.2, 262.2, 249.2, 236.2 K |
| CTH (ice clouds) | 4, 6, 8, 10, 12 km |
| CTT* (ice clouds) | 262.2, 249.2, 236.2, 223.3, 216.7 K |
| geometric thickness | 1 km |
| cloud particle scattering | on / off |

* corresponds to CTH

| setup of atmosphere, geometry and surface | |
|---|---|
| atmosphere | US-standard |
| molecular absorption | on / off |
| SATZ | 0° |
| SKT | 290 K |
| surface type | ocean |

and is used to compute BTs from measured radiances in remote sensing.

The simplest version of the BTD Nonlinearity Shift can be explained using the Schwarzschild equation for radiative transfer. The Schwarzschild equation is a simple version of radiative transfer assuming no cloud scattering and no atmosphere. Its solution for one cloud layer is

$$R^{\text{S}}_{\text{TOA},\lambda}(\tau_\lambda) = e^{-\tau_\lambda} B_\lambda(T_s) + (1 - e^{-\tau_\lambda}) B_\lambda(\text{CTT}), \tag{6}$$





where $R_{\mathrm{TOA},\lambda}^{\mathrm{S}}$ is the radiance at TOA at a given wavelength $\lambda$ with the superscript S for Schwarzschild, and $\tau_\lambda$ is the optical thickness of the cloud for this wavelength. The first term in the equation is the transmitted radiance coming from the surface with the surface temperature $T_s$; the second term is the radiation emitted by the cloud, assuming that the cloud layer has an approximately constant temperature $T \approx \mathrm{CTT}$. To demonstrate the BTD Nonlinearity Shift we set $\tau_\lambda$ equal for all wavelengths, $\tau_\lambda = \tau$. Figure 3(a) shows the Planck function of the surface temperature, $B_\lambda(T_s)$, and the cloud temperature, $B_\lambda(\mathrm{CTT})$, in grey for exemplary values of $T_s = 290\,K$ and CTT$= 200\,K$. According to the Schwarzschild equation (Eq. 6), $R_{\mathrm{TOA},\lambda}^{\mathrm{S}}$ lies between these two curves, approaching $B_\lambda(T_s)$ for $\tau \to 0$ and $B_\lambda(\mathrm{CTT})$ for $\tau \to \infty$. Figure 3(a) illustrates $R_{\mathrm{TOA},\lambda}^{\mathrm{S}}$ for $\tau = 0.5$ (black line). From $R_{\mathrm{TOA},\lambda}^{\mathrm{S}}$ we can now compute the TOA BTs at the three IR wavelengths of interest as $\mathrm{BT}_\lambda^{\mathrm{S}} = T_\lambda(R_{\mathrm{TOA},\lambda}^{\mathrm{S}}(\tau))$, where the superscript S again stands for Schwarzschild. The corresponding Planck curves, i.e. $B_\lambda(\mathrm{BT}_\lambda^{\mathrm{S}})$ for $\lambda \in \{8.7, 10.8, 12.0\}$, are shown in Fig. 3(a) as dashed colored lines. Recall that in this example calculation we have set a constant $\tau = 0.5$, i.e. the same optical properties (transmittance and emissivity) for all wavelengths (see Eq. 6). Naively, one might expect a BTD = 0 (i.e. equal BTs) in this scenario. However, it is evident from the figure that the three BTs are different, with $\mathrm{BT}_{8.7}^{\mathrm{S}} > \mathrm{BT}_{10.8}^{\mathrm{S}} > \mathrm{BT}_{12.0}^{\mathrm{S}}$. Since these differences between the three BTs are not due to optical cloud properties, they must be caused by the nonlinear transformation from radiances to BTs. Hence, the BTD Nonlinearity Shift induces a BTD in situations where, naively, no BTD would be expected.

To get an overview of the BTD Nonlinearity Shift, we compute BTD$^{\mathrm{S}}$ for both wavelength combinations (BTD(8.7-10.8) and BTD(10.8-12.0)) from the results of the Schwarzschild equation (Eq. 6) for varying $\tau$ and CTT as

$$\mathrm{BTD}^{\mathrm{S}}(\lambda_0 - \lambda_1) = T_{\lambda_0}(R_{\mathrm{TOA},\lambda_0}^{\mathrm{S}}(\tau)) - T_{\lambda_1}(R_{\mathrm{TOA},\lambda_1}^{\mathrm{S}}(\tau)) \,. \tag{7}$$

Fig. 3(b) shows the computed BTD$^{\mathrm{S}}$ as a function of $\tau$ for different CTTs and a fixed $T_s$=290 K. These BTD$^{\mathrm{S}}$ resemble an arc shape (similar to the well-known BTD arc from Inoue (1985)) and show higher values for lower CTTs, even though the amplitudes of their curves are smaller than for the full RT model, as we will see later. Thus, even if $\tau_\lambda$ is the same for all three wavelengths, $\tau_\lambda = \tau$, the nonlinearity of the inverse Planck function induces positive BTD$^{\mathrm{S}}$ values and a dependence on the CTT. Notice that for these examples the BTD induced this way reaches up to 2.5 K and thus cannot be neglected.

In the next section we will discuss the effects of cloud properties on the BTDs due to the wavelength-dependent optical properties in the full RT model (described in Sect. 3). The BTD Nonlinearity Shift adds to these effects and is therefore co-responsible for the (positive) BTD values and the CTT dependence of the BTDs which we will discuss in more detail in Sect. 5.6. In appendix B we further analyse the BTD Nonlinearity Shift for the Schwarzschild model as well as the full RT model and disentangle this nonlinearity effect from the physical effects of wavelength-dependent optical properties on the BTDs in RT calculations.

Summarizing this section:

– There is an effect (BTD Nonlinearity Shift) coming from the nonlinearity of the inverse Planck function that induces positive BTD(8.7-10.8) and BTD(10.8-12.0) values and a dependence on the CTT in a simple RT model (Schwarzschild equation) even if cloud optical properties (transmittance and emissivity) are the same for all wavelengths.





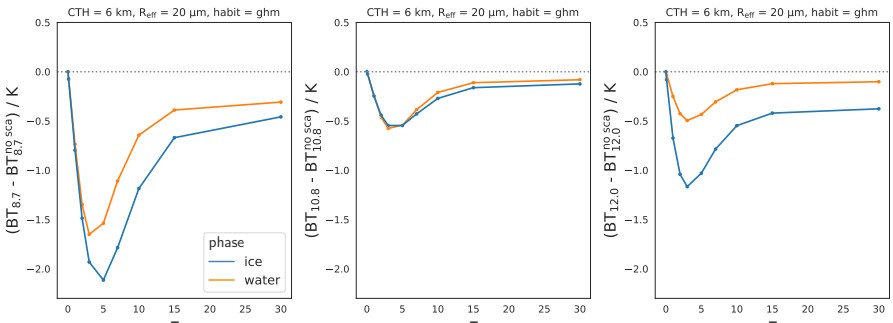

**Figure 4.** Scattering effects on brightness temperatures (BT): Difference between the BTs for a cloud with scattering and a cloud with scattering switched off for all three IR window channels, i.e. $BT_\lambda - BT_\lambda^{\mathrm{no\,sca}}$ for each channel with centre wavelength $\lambda \in \{8.7\mu m, 10.8\mu m, 12.0\mu m\}$, for liquid and ice clouds as functions of optical thickness $\tau$.

## 5 Effects of cloud properties on BTDs

In this section we analyse the results of the RT calculations described in Sect. 3. We start with the effects of scattering on the
230  BTs of the three window channels separately. We then combine the BTs to BTDs and analyse them as functions of $\tau$, phase, $R_{\mathrm{eff}}$, ice crystal habit and CTT, focusing on the physical relationships between these cloud properties and the BTDs. In order to disentangle the effects of the different cloud parameters, we always vary only one or two parameters and keep the remaining cloud parameters at fixed "default" values, namely CTH=6 km (corresponding to CTT=249.2 K), $R_{\mathrm{eff}}$=20 $\mu$m for both cloud phases and the general habit mix as ice crystal habit. Afterwards we combine all the dependencies to perform a sensitivity
235  analysis of the BTDs with respect to the cloud parameters in Sect. 6.

The following conventions are used throughout this section: blue colours indicate the ice phase; orange/red colours indicate the liquid phase. Solid lines represent a 'normal' atmosphere with molecular absorption; dashed lines mean that molecular absorption is switched off.

### 5.1 Effects of scattering on brightness temperatures

240  Scattering in the infrared window only needs to be considered for cloud particles; Rayleigh scattering by atmospheric molecules is negligible in the infrared window. The effects of cloud particle scattering on the BTs is shown in Fig. 4. It shows the difference between the BTs for a cloud with scattering and a cloud with scattering switched off for the three window channels, i.e. $BT_\lambda - BT_\lambda^{\mathrm{no\,sca}}$ for each channel with centre wavelength $\lambda$. This is shown as a function of $\tau$ (at 550 nm) for an ice and a water cloud with all other cloud parameters held constant. Switching off scattering in a cloud changes the optical thickness of that
245  cloud, since only absorption now contributes to the extinction of radiation. However, to be able to compare scenarios with and without scattering, the $\tau$ parameter used for this figure is still the "original" optical thickness (with absorption and scattering).





This means that for a given $\tau$ in the figures the water content is held constant for the scenario with and without scattering. Thus, Fig. 4 compares radiances between a cloud with and without scattering, but for fixed microphysical parameters.

All curves in Fig. 4 are negative everywhere, meaning that scattering is a radiation sink for all three wavelengths: Part of the radiation coming from below the cloud is scattered back downwards. However, the amount of radiation lost to scattering is different for the different wavelengths. Scattering has a larger effect on the radiation at $8.7\,\mu$m than at $10.8$ or $12.0\,\mu$m, as expected from $\beta_{\mathrm{sca}}$ which is higher at $8.7\,\mu$m. For $8.7$ and $12.0\,\mu$m, scattering by ice clouds is more significant than by water clouds; for $10.8\,\mu$m, scattering leads to a similar radiation loss for both water and ice clouds. Interestingly, scattering effects are visible even when the cloud is opaque (black, $\tau$=30). An explanation is that the observed radiance at TOA does not just come from the top of the cloud. Rather, it comes from the upper layers within the cloud (with decreasing intensity as one moves deeper into the cloud). Radiation emitted anywhere below the cloud top is still subject to scattering on its way to the cloud top.

### 5.2 Effects of optical thickness on BTDs

We begin the study of BTDs by analysing the physical factors that drive the BTDs' behavior in relation to $\tau$. Fig. 5 shows BTD(8.7-10.8) and BTD(10.8-12.0) as functions of $\tau$ for both an ice and a liquid cloud and with molecular absorption switched on and off. To disentangle the effects of cloud absorption and scattering, Fig. 5(a,b) show the BTDs with cloud particle scattering switched off. As explained in the previous section, the $\tau$ parameter used for these figures is still the "original" optical thickness (with absorption and scattering). In Fig. 5(c,d) scattering is switched on.

As $\tau$ approaches zero in all panels of Fig. 5, i.e. no cloud is simulated, the BTD curves with atmospheric absorption switched on (solid lines) do not go to zero. They remain above zero for BTD(10.8-12.0) and below zero for BTD(8.7-10.8). This is the effect of atmospheric absorption, since radiation at $8.7\mu$m and $12.0\,\mu$m is more strongly absorbed by water vapour than at $10.8\mu$m: Compare the curves with (solid lines) and without (dashed lines) molecular absorption for $\tau$ approaching zero. As $\tau$ increases, the curved shape of the BTD functions is (largely) due to the interplay of transmission and emission from the cloud. As discussed in Sect. 4 the BTD Nonlinearity Shift adds to these effects. The amount of transmitted radiation is different at different wavelengths because extinction by cloud particles (i.e. the combined effects of absorption and scattering that remove radiation from the beam in a given direction) is wavelength dependent. This results in non-zero BTD values where transmission is dominant (small $\tau$). As the cloud becomes optically thicker, the emission from the cloud becomes more important than the transmitted radiation. As the emission is similar at the three wavelengths, the BTD values are small where the emission is the dominant radiation source (large $\tau$), giving rise to the curved shape of the BTD functions (the well-known BTD arc from Inoue (1985)). The BTD curves become constant at about $\tau \gtrsim 15$.

Fig. 5(a) shows BTD(10.8-12.0) without cloud particle scattering, i.e. only the effects of absorption and emission in the cloud are visible. This BTD is positive, meaning that radiation at a wavelength of $12.0\,\mu$m is more strongly absorbed than at $10.8\,\mu$m and more radiation is transmitted through the cloud at $10.8\,\mu$m. This matches the absorption coefficient, which is higher at $12.0$ than $10.8\,\mu$m (shown as an inset for the given $R_{\mathrm{eff}}$ for convenience, as well as in Fig. 2).

Analogously, Fig. 5(b) shows that radiation at $10.8\,\mu$m is more strongly absorbed by the cloud than at $8.7\,\mu$m, especially for ice clouds. The stronger absorption at $10.8$ compared to $8.7\,\mu$m can again be seen in the absorption coefficient (shown in inset





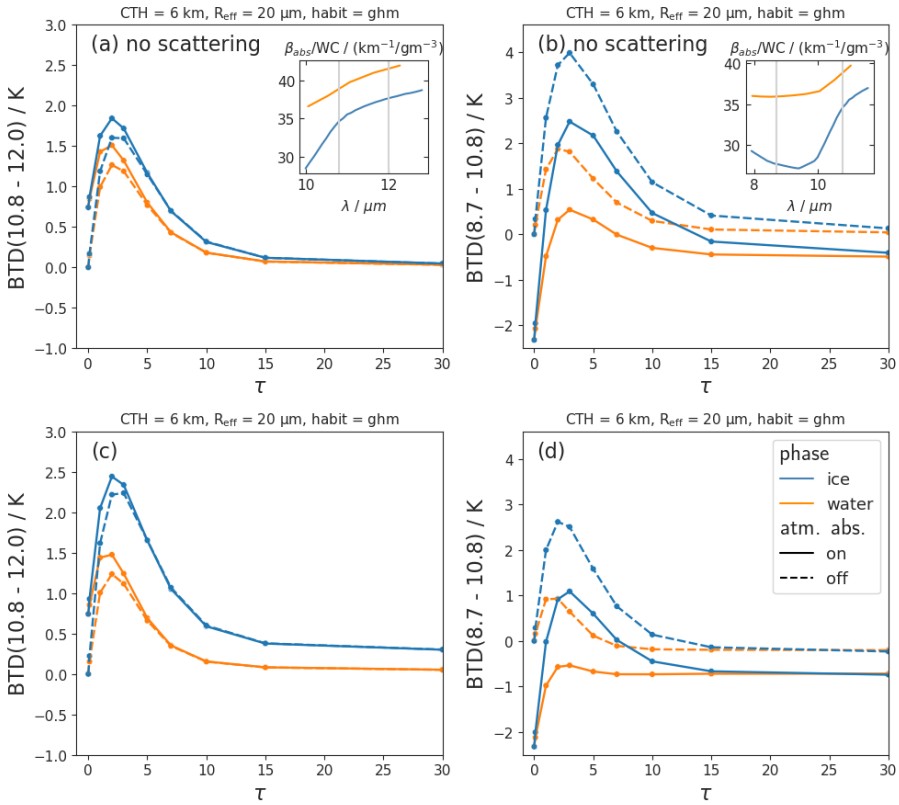

**Figure 5.** Brightness temperature differences BTD(10.8-12.0) and BTD(8.7-10.8) as functions of $\tau$ for cloud particle scattering (a,b) switched off and (c,d) switched on for liquid and ice clouds. Solid lines indicate a "normal" absorbing atmosphere, dashed lines indicate that molecular absorption is switched off.

and in Fig. 2). The spectral differences in the absorption coefficient are stronger between 8.7 and 10.8 than between 10.8 and 12.0 $\mu$m, leading to higher values of BTD(8.7-10.8) than BTD(10.8-12.0) (compare Fig. 5(a) to Fig. 5(b)). For BTD(8.7-10.8), note that molecular absorption plays an important role even for optically thick clouds, decreasing BTD(8.7-10.8) everywhere by at least 0.5 K, since radiation at 8.7 $\mu$m is more strongly absorbed by atmospheric molecules (water vapour) than at 10.8 $\mu$m.

Switching on particle scattering (Fig. 5(c)), the BTD(10.8-12.0) values increase for ice clouds and stay about the same for liquid clouds. This will be further discussed in the next section (Sect. 5.3). For opaque clouds (large $\tau$), the spectral differences in scattering effects lead to non-vanishing BTD(10.8-12.0) values for ice clouds (BTD(10.8-12.0) $\approx 0.3$ K).

We saw that the positive values of BTD(8.7-10.8) when scattering is switched off are due to weaker absorption at 8.7 than 10.8$\mu$m. Scattering is on the other hand stronger at 8.7 compared to 10.8 $\mu$m (see Fig. 4). This leads to a decrease in BTD(8.7-290    10.8) when scattering is switched on (greater decrease for $BT_{8.7}$ compared to $BT_{10.8}$; compare Fig. 5(b) with Fig. 5(d)) – an effect on BTD(8.7-10.8) opposite to that of absorption. However, the decrease due to cloud scattering is not strong enough to overcome the increase in BTD(8.7-10.8) due to cloud absorption and the BTD(8.7-10.8) curve is still positive (when



atmospheric absorption is not considered). Note the differences with BTD(10.8-12.0), where cloud absorption and scattering are concurrent effects, both leading to an increase in BTD(10.8-12.0).

The following list summarizes the most important results:

– Stronger absorption and scattering at 12.0 compared to 10.8 $\mu$m lead to positive values of BTD(10.8-12.0).

– Stronger absorption at 10.8 compared to 8.7 $\mu$m lead to positive values of BTD(8.7-10.8); scattering has a mediating effect, reducing BTD(8.7-10.8) values.

– These trends are consistent with expectations based on absorption and scattering coefficients.

**5.3    Effects of cloud phase on BTDs**

We now discuss the direct dependence of BTD(10.8-12.0) and BTD(8.7-10.8) on phase shown in Fig. 5. Direct dependence means that all other parameters such as $R_{\mathrm{eff}}$ or CTT are held constant. BTD(10.8-12.0) in Fig. 5(c) has higher values for the ice phase than the liquid phase for all $\tau$. Comparing the curves with and without scattering (Fig. 5(a) and Fig. 5(c)), we see that this difference between liquid and ice is mainly due to the different scattering properties of cloud particles at the two wavelengths:
For liquid clouds the scattering has a similar effect at 10.8 and 12.0 $\mu$m, while for ice clouds radiation at 12.0 $\mu$m is scattered more than at 10.8 $\mu$m (see Fig. 4), leading to higher BTD(10.8-12.0) values for ice clouds.

BTD(8.7-10.8) directly depends on phase only for small to moderate $\tau$ ($\tau \lesssim 15$), with higher values for ice than for liquid. This difference is due to absorption properties: The spectral difference in absorption between the two wavelengths is larger for ice clouds (see $\beta_{\mathrm{abs}}$ in the inset of Fig. 5(b) or Fig. 2). Switching on cloud scattering reduces the differences between ice
and liquid clouds in BTD(8.7-10.8) (compare Fig. 5(b) with Fig. 5(d)). The reason for this can be seen in Fig. 4: The effect of scattering at 8.7 $\mu$m is stronger for ice than for water, while it is similar for ice and for water at 10.8 $\mu$m. This leads to a stronger decrease in BTD(8.7-10.8) values for ice than for water clouds when scattering is switched on. However, overall the effect of absorption (leading to larger BTD(8.7-10.8) values for ice than for water) outweighs this contrasting scattering effect.

In summary, the most important findings are:

– There is a direct phase dependence of the BTDs due to the dependence of the single scattering properties on cloud phase.

– This effect is of the order of 0.5–1.5 K for BTD(10.8-12.0) and 0–2 K for BTD(8.7-10.8), depending on $\tau$, in all modeled scenarios.

– For BTD(10.8-12.0), mainly scattering is responsible for the direct dependence on cloud phase.

– For BTD(8.7-10.8), absorption is responsible for the direct dependence on cloud phase, scattering reduces the differences
between the phases.





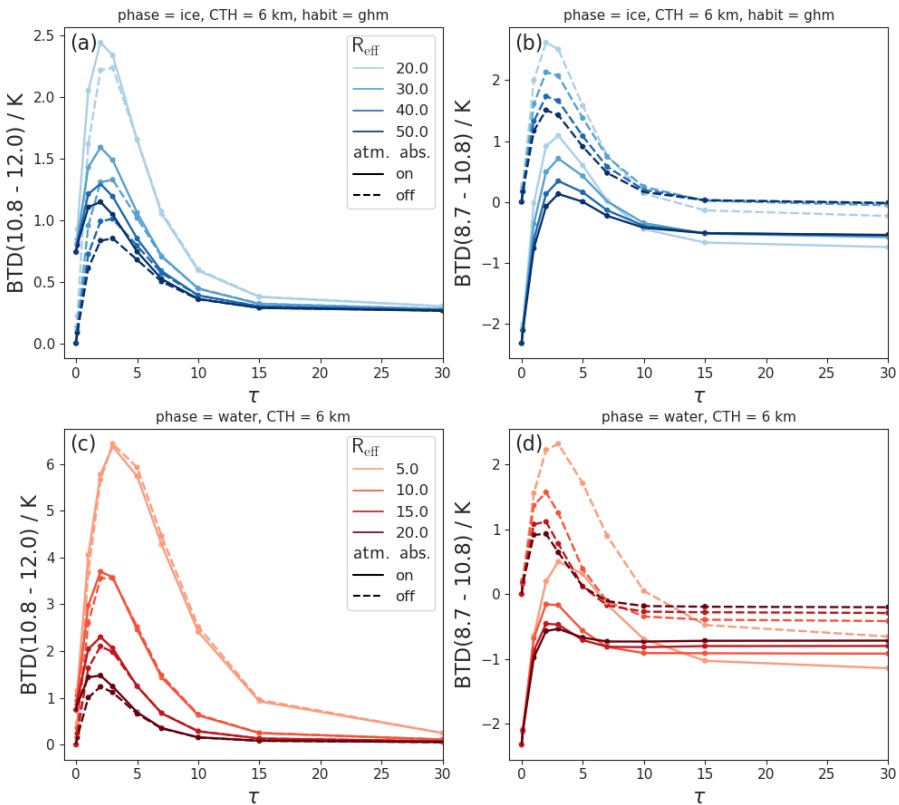

**Figure 6.** Effects of varying $R_{eff}$ on BTD(10.8-12.0) and BTD(8.7-10.8) as functions of $\tau$ for ice clouds (top row) and liquid clouds (bottom row). Solid lines indicate a "normal" absorbing atmosphere, dashed lines indicate that molecular absorption is switched off.

## 5.4 Effects of effective radius on BTDs

Fig. 6 shows BTD(10.8-12.0) and BTD(8.7-10.8) as a function of $\tau$ and $R_{eff}$ for ice clouds (top row) and liquid clouds (bottom row) for the full RT model (i.e. scattering switched on). Note that the range of $R_{eff}$ values for ice and liquid clouds are different in order to simulate realistic cloud conditions. For low $\tau$ ($\tau \lesssim 10$), smaller $R_{eff}$ lead to larger values for both BTDs. The effect

becomes stronger in a nonlinear way as the $R_{eff}$ becomes smaller. This confirms previous results, for instance Dubuisson et al. (2008), who also found a strong and nonlinear dependence of BTDs on $R_{eff}$.

The effect of $R_{eff}$ on BTD(10.8-12.0) results physically from the dependence of particle absorption on $R_{eff}$: The spectral differences of the absorption coefficient are larger for smaller $R_{eff}$ (see Fig. 2), resulting in lower transmission at 12.0 than at $10.8\,\mu m$, and thus higher BTD(10.8-12.0) values for smaller $R_{eff}$ values. The effect of scattering on BTD(10.8-12.0) is

similar for varying $R_{eff}$ and comparatively small (increases (decreases) the BTD by $\lesssim 0.5\,K$ for ice (water) clouds). For the interested reader, Fig. C1 in appendix C shows the sensitivity of both BTDs with $R_{eff}$ broken down into effects of absorption and scattering.



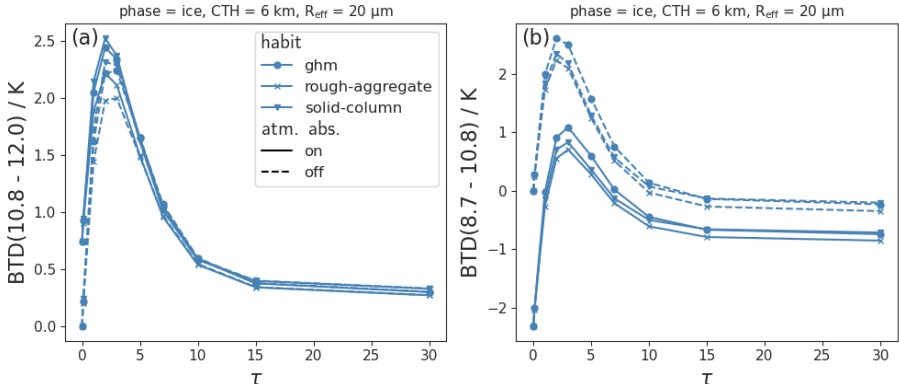

**Figure 7.** Effects of varying ice crystal habit on (a) BTD(10.8-12.0) and (b) BTD(8.7-10.8) as functions of $\tau$ for ice clouds. Solid lines indicate a "normal" absorbing atmosphere, dashed lines indicate that molecular absorption is switched off.

For BTD(8.7-10.8), the $R_{eff}$ dependence for small $\tau$ is, like the phase dependence, the result of two opposite effects: For smaller $R_{eff}$, absorption increases for 10.8 compared to 8.7 $\mu$m, leading to an increase in BTD(8.7-10.8). On the other hand
scattering increases more for 8.7 than for 10.8 $\mu$m, leading to a decrease in BTD(8.7-10.8). However, the effect due to absorption is stronger and therefore the BTD(8.7-10.8) increases with decreasing $R_{eff}$. Unlike BTD(10.8-12.0), BTD(8.7-10.8) is still dependent on $R_{eff}$ at large $\tau$: here BTD(8.7-10.8) increases with increasing $R_{eff}$, contrary to the $R_{eff}$ trend at small $\tau$. The smaller the $R_{eff}$, the more important this effect becomes.

Summarizing the most important insights:

– The BTDs depend strongly and nonlinearly on $R_{eff}$.

– Physically this dependence is due to larger spectral differences in the absorption coefficient for smaller $R_{eff}$.

– For BTD(8.7-10.8), stronger scattering for smaller $R_{eff}$ mediates the absorption effects.

**5.5   Effects of ice crystal habit on BTDs**

Figure 7 shows the sensitivity of the BTDs on ice crystal habits (in ice clouds). For both BTDs, rough aggregates lead to the
smallest BTD values. For BTD(8.7-10.8), ice crystals with the general habit mix (ghm) lead to the largest BTD values, while for BTD(10.8-12.0), solid columns lead to slightly higher values. However, the sensitivity on ice crystal habits is relatively small ($\lesssim 0.5$ K) compared to other cloud properties. This confirms Dubuisson et al. (2008), who showed that the habit has a small effect on BTDs compared to the effect of $R_{eff}$ also for other ice crystal shapes than the ones considered here. The relative importance of different cloud parameters will be further discussed in Sect. 6.



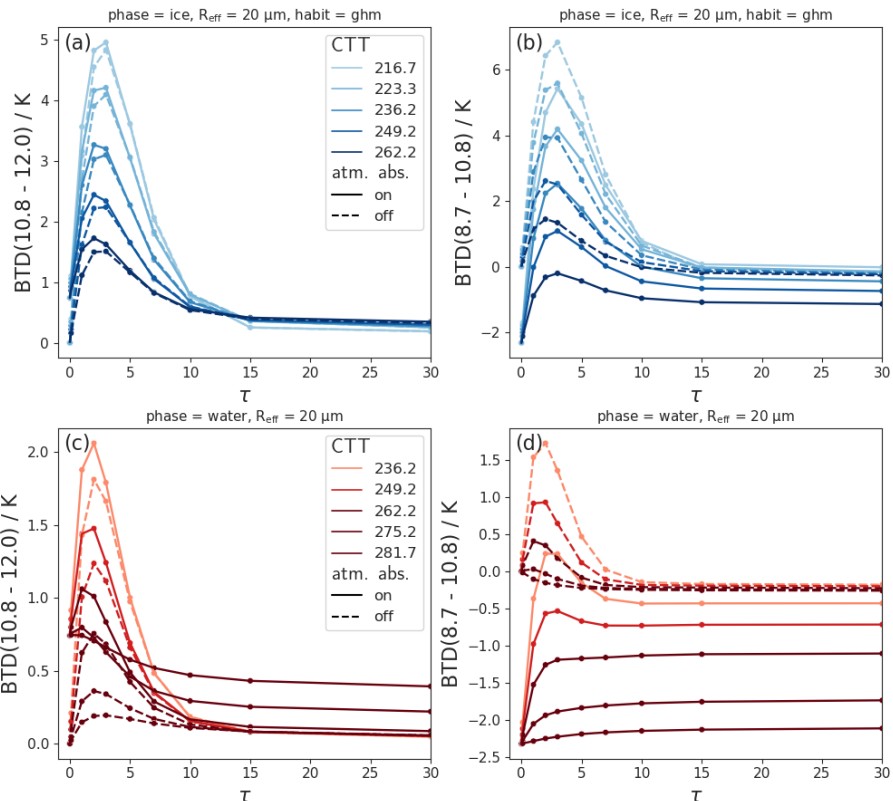

**Figure 8.** Effects of varying cloud top temperature (CTT) on BTD(10.8-12.0) and BTD(8.7-10.8) as a function of $\tau$ for ice clouds (top row) and liquid clouds (bottom row). Solid lines indicate a "normal" absorbing atmosphere, dashed lines indicate that molecular absorption is switched off.

## 5.6 Effects of cloud top temperature on BTDs

Figure 8 shows the sensitivity of both BTDs to CTT – and thus to CTH – for ice (top row) and liquid (bottom row) clouds. The results with molecular absorption switched off (dashed lines) show how much of this sensitivity is due to the atmosphere. Note that the CTT ranges for ice and liquid clouds are different in order to simulate realistic cloud conditions.

For BTD(10.8-12.0), molecular absorption is only relevant for small $\tau$. For BTD(8.7-10.8), however, molecular absorption is relevant for all $\tau$ values: Clouds with high CTT, i.e. low CTH, have more absorbing atmosphere above cloud top, leading to more radiation absorbed at 8.7 compared to 10.8 $\mu$m.

At low $\tau$ ($\lesssim 10$), both BTD(8.7-10.8) and BTD(10.8-12.0) show a strong dependence on CTT that is not due to molecular absorption. This CTT effect is also not due to differences in absorption or scattering, since the single scattering properties are not CTT dependent (see Sect. 2). Similarly, cloud emissivity is not CTT dependent (as it is a function of the absorption coefficient).





In Sect. 4 we found that the BTD Nonlinearity shift leads to a CTT dependence of the BTDs with higher BTD values for lower CTTs even when optical cloud properties are the same for all wavelengths. This explains part of the CTT dependence in Fig. 8. In appendix B we further discuss the BTD Nonlinearity Shift, allowing also wavelength dependent optical properties. It can be shown that for the Schwarzschild $BTD^S$, spectral differences in the extinction coefficient are scaled by the difference

between the surface and the cloud top radiance, $B_\lambda(T_s) - B_\lambda(CTT)$ (see appendix B for a detailed discussion). Hence, the effects of spectral differences in optical properties on $BTD^S$ are amplified by larger differences between $T_s$ and the CTT. This is the main reason (besides the BTD Nonlinearity Shift) for the CTT dependence of the BTDs. Colder CTTs thus increase both the BTD Nonlinearity Shift and the effects of spectral differences in optical properties.

The following list summarizes the CTT / CTH effects on the BTDs:

– For BTD(8.7-10.8), CTH has a large effect, due to molecular absorption mainly above cloud top.

   – Both BTDs show a strong CTT dependence with higher values for lower CTTs.

   – The BTD Nonlinearity Shift is co-responsible for the positive BTD values and the CTT dependence of the BTDs, adding to the effects stemming from spectral differences in absorption and scattering properties.

## 6 Implications for phase retrievals

In this section we assess the sensitivity of the BTDs in a standard atmosphere by combining the four phase related cloud parameters $\tau$, $R_{eff}$, CTT and thermodynamic phase. From this sensitivity analysis we understand the relative importance of the different cloud parameters and which cloud parameters are responsible for the phase information contained in the BTDs. This allows us to derive implications for phase retrievals. First, we perform sensitivity analyses for each BTD individually, examining the phase information content of each BTD. Next, we study the sensitivities and phase information content of the

two BTDs combined.

### 6.1 Sensitivity analysis for each BTD

Fig. 9 gives an overview of the sensitivities of the BTDs for "typical" cloud scenarios, as defined in the following. The figure shows the BTDs for upper and lower boundaries of CTT (217 – 249 K for ice, 262 – 282 K for liquid water) and $R_{eff}$ (20 – 50 $\mu$m for ice, 5 – 20 $\mu$m for liquid water). These ranges are representative for mid-latitude clouds (between 30 and 50°N or S)

and are chosen as follows: The CTT boundaries are derived from the active remote sensing product DARDAR (liDAR/raDAR, Delanoë and Hogan (2010)) - specifically, values close to the 15th and 85th percentiles of ice and liquid CTTs observed for mid-latitude clouds, covering about 70% of CTTs (see Mayer et al. (2023) for detailed information on the data set). The cloud scenarios with the two CTT boundary values per phase are shown in different colors in Fig. 9 (light blue and dark blue for ice clouds; orange and red for liquid clouds). For the $R_{eff}$ boundaries we select the upper and lower limits of all computed

$R_{eff}$ scenarios (see table 1). Additionally, as liquid clouds rarely have $\tau < 5$, these values are omitted, since we focus for this sensitivity analysis on "typical" cloud scenarios. For ice clouds, different habits are shown as different markers. Hence, the



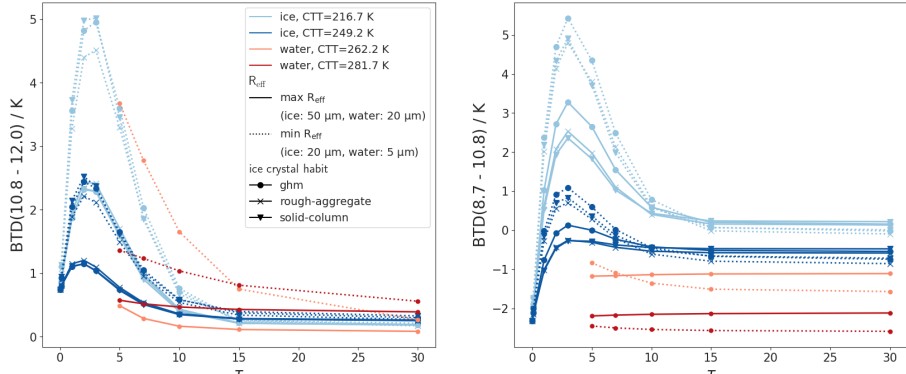

**Figure 9.** Sensitivity analysis for each BTD: BTD(10.8-12.0) and BTD(8.7-10.8) for typical upper and lower boundaries of CTT and $R_{eff}$ for ice (blue colors) and liquid (orange/ red colors) clouds. For ice clouds, different habits are shown as different markers.

cloud parameters in Fig. 9 are chosen such that the majority of mid-latitude cloud events for each phase lie between the very bottom and top blue curves for ice and the very bottom and top orange/red curves for liquid.

To verify that the computed ranges of BTD values are realistic, we compare the RT results with measured SEVIRI data using
cloud phase information from DARDAR. More details on this comparison and its results can be found in Appendix D. We find good agreement between the RT results and the measured SEVIRI data and conclude that the results of the RT calculations are realistic.

Starting with BTD(10.8-12.0), the sensitivity plot in Fig. 9 reveals the following: For small $\tau$, BTD(10.8-12.0) shows high sensitivity to $\tau$, $R_{eff}$ and CTT. The sensitivity to $R_{eff}$ is nonlinear (see Sect. 5.4) and dependent on CTT: the sensitivity to $R_{eff}$
is stronger for smaller $R_{eff}$ and lower CTTs. This dependence on CTT is explained in Sect. 4, where we found that the effect of spectral differences in the cloud optical properties (which are larger for smaller $R_{eff}$; see Fig. 2) is amplified by the temperature difference between the surface and the cloud top (see also Eq. B7 and Fig. B1 in the appendix). As $\tau$ increases, all curves tend to group together and show a comparatively small dependence on CTT and $R_{eff}$. Recall that the direct phase dependence of BTD(10.8-12.0), i.e. holding all other cloud parameters constant, is of the order of 0.5–1.5 K (see Sect. 5.3). Hence, the direct
sensitivity to cloud phase is mostly smaller than the sensitivity to CTT and $R_{eff}$. Furthermore, the sensitivity to ice crystal habit is relatively modest ($\lesssim 0.5$ K in all cases). The largest BTD(10.8-12.0) values in these "typical" cloud scenarios (about 2.5 to 5 K in the figure) are only observed for optically thin and cold ice clouds with small $R_{eff}$. Thus BTD(10.8-12.0) is useful to detect cirrus clouds, especially if they have small $R_{eff}$ (like contrails), and classify them as ice in a phase retrieval. However, our calculations show that certain liquid cloud scenarios with exceptionally low $R_{eff}$ and cold CTTs can also give remarkably
high BTD(10.8-12.0) values. This can lead to misclassification of these liquid clouds as ice. However, most liquid clouds have lower BTD(10.8-12.0) values (below about 2.5 K in the figure). Since such low BTD(10.8-12.0) values may also indicate ice clouds with warm CTTs and/or large $R_{eff}$, or ice clouds with $\tau$ close to zero, a phase classification based on BTD(10.8-12.0)



alone is challenging. The lowest BTD(10.8-12.0) values (about 0 to 1 K in the figure) indicate optically thick clouds, but do otherwise not contain much phase information.

Like BTD(10.8-12.0), BTD(8.7-10.8) is very sensitive to $\tau$ and CTT. Since BTD(8.7-10.8) is strongly dependent on molecular absorption (see Sect. 5.6), the sensitivity to CTT is not only relevant for small $\tau$ (as for BTD(10.8-12.0)), but also for large $\tau$. At large $\tau$, sensitivity to CTT reflects the influence of CTH and thus the atmospheric column above the cloud top. This results in generally lower BTD(8.7-10.8) values for liquid clouds which typically have lower CTHs. In comparison to $\tau$ and CTT/CTH the sensitivity to $R_{\text{eff}}$ is lower and mainly relevant for small CTT. Interestingly, BTD(8.7-10.8) is also sensitive

to $R_{\text{eff}}$ for opaque clouds ($\tau = 30$), as mentioned in Sect. 5.4. Recall from Sect. 5.3 that the direct influence of cloud phase (holding other parameters constant) on BTD(8.7-10.8) plays a significant role only for small $\tau$ values (around $\lesssim 10$). For larger $\tau$ values the direct influence of the phase is negligible and the separation between ice and liquid clouds in the figure is mainly due to differences in CTH and to a lesser extent due to differences in $R_{\text{eff}}$. For $\tau \lesssim 10$, the direct phase effect on BTD is of the order of 1–2 K. Hence, the direct influence of phase on BTD(8.7-10.8) for small $\tau$ is smaller than the sensitivity to CTT,

as is the case for BTD(10.8-12.0). The sensitivity of BTD(8.7-10.8) to ice crystal habit is larger than for BTD(10.8-12.0) and increases for lower CTTs ($\approx 1$ K for the coldest shown CTT and $R_{\text{eff}} = 50\,\mu\text{m}$, $< 0.5$ K in all other cases). It is however again less important than the sensitivity to the other cloud parameters. The BTD(8.7-10.8) peaks (around 1 to 5.5 K in Fig. 9) are associated with ice clouds with low $\tau$ of about $1 < \tau < 7$. Thus, large BTD(8.7-10.8) values can indicate ice phase. The interpretation of low BTD(8.7-10.8) values (about $-2$ to 0 in the figure) is more complex: This range of values can come from

very thin ice clouds (as BTD(8.7-10.8) decreases to about -2 as $\tau$ goes to zero), optically thick ice clouds, or liquid clouds with cold CTTs. For optically thick clouds, BTD(8.7-10.8) values decrease with higher CTT (due to lower CTHs and stronger molecular absorption) and smaller $R_{\text{eff}}$ - both characteristics typical of liquid clouds. As a general guideline for optically thick clouds, lower BTD(8.7-10.8) values indicate a higher probability of a liquid cloud. The $R_{\text{eff}}$ dependence of BTD(8.7-10.8) for large $\tau$ can be valuable in determining the phase of clouds with large $\tau$, where lower $R_{\text{eff}}$ values indicate a larger liquid cloud

probability. Values of BTD(8.7-10.8) below a certain threshold (depending on molecular absorption; about $-2$ K in the figure) always indicate a liquid cloud.

Overall, the phase information contained in BTD(8.7-10.8) comes mainly from its sensitivity to CTT for clouds with $\tau \lesssim 10$ and from its sensitivity to CTH and (to a lesser extent) $R_{\text{eff}}$ for optically thick clouds. Only in cases of optically thin liquid clouds ($\tau \lesssim 10$) is the phase information of BTD(8.7-10.8) additionally due to the direct phase influence on the absorption

properties of liquid and ice particles.

To summarize the main findings:

– The sensitivities of the BTDs are complex.

– BTD(10.8-12.0) shows the highest sensitivity to $\tau$, CTT and $R_{\text{eff}}$. BTD(8.7-10.8) shows the highest sensitivity to $\tau$ and CTT/CTH.

– Thin ice clouds can be detected by both BTD(10.8-12.0) and BTD(8.7-10.8) as long as $\tau \gtrsim 1$.



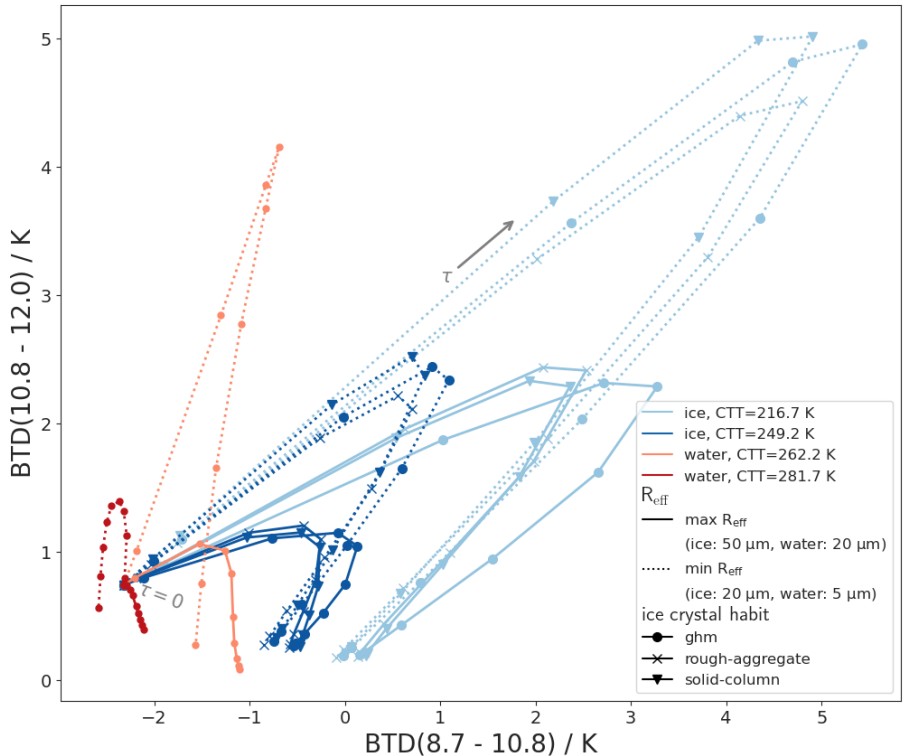

**Figure 10.** Sensitivity analysis for the combination of both BTDs: Blue lines show ice clouds; orange/red lines show liquid clouds for typical upper and lower boundaries of CTT and $R_{eff}$. Along each line, $\tau$ increases from 0 to 30. For ice clouds, different habits are shown as different markers

– BTD(8.7-10.8) also provides CTH and $R_{eff}$ information for optically thick clouds, which can be useful for phase determination.

– For BTD(10.8-12.0), typical liquid and ice clouds overlap for most cloud scenarios, with the exception of cold, thin ice clouds. For BTD(8.7-10.8), liquid and ice clouds separate better, but the BTD values of the two phases are close when CTTs (CTHs) are similar. This phase separation is mainly due to the sensitivity of BTD(8.7-10.8) to CTT/ CTH.

### 6.2 Sensitivity analysis for the combination of both BTDs

We perform a similar sensitivity analysis as in the last section for the combination of both BTDs. As Fig. 9, Fig. 10 shows the BTDs for the same upper and lower boundaries of CTT and $R_{eff}$, but in the space spanned by BTD(8.7-10.8) and BTD(10.8-12.0). Along each line, $\tau$ increases from 0 to 30. To make the shape of the curves easier to understand, here also liquid clouds with $\tau < 5$ are shown (in contrast to Fig. 9).





**Figure 11.** Left column: BTD(10.8-12.0) and BTD(8.7-10.8) values within the defined "typical" boundaries of CTT. Round markers indicate liquid clouds; crosses indicate ice clouds. Clouds which can be distinguished using a CTT proxy like $BT_{10.8}$, i.e. optically thick clouds ($\tau \geq 10$) with very low ($< 233\,\mathrm{K}$) or very high ($> 273\,\mathrm{K}$) CTTs, are not shown. The color code in the three rows encodes $\tau$, CTT and $R_{\mathrm{eff}}$ respectively. Right column: Same as left column but for the whole range of computed cloud scenarios (see table 1), including also exceptionally cold liquid clouds and exceptionally warm ice clouds. The color codes of the three rows correspond to the color codes in the left column. Clouds which can be distinguished using a CTT proxy are again not shown.



Figure 10 shows that the combined knowledge of both BTD(8.7-10.8) and BTD(10.8-12.0) leads to a better phase classification than considering BTD(8.7-10.8) and BTD(10.8-12.0) individually. For instance, liquid clouds at cold CTTs and small $R_{eff}$ (orange dotted line) separate from ice clouds in Fig. 10 as long as $\tau$ is not too large ($\lesssim 10$). In contrast, the same cloud scenario overlaps with ice cloud scenarios when only BTD(8.7-10.8) or BTD(10.8-12.0) are considered individually (Fig. 9).

In order to better showcase the range of BTD values for both phases and identify overlap regions, we use an additional type of plot: Instead of showing only the boundary cases (as in Fig. 10), the left column of Fig. 11 shows (almost) all computed BTD values within the defined boundaries of CTT and $R_{eff}$ in the space spanned by the two BTDs. Only optically thick clouds ($\tau \geq 10$) with very low ($< 233\,\mathrm{K}$) or very high ($> 273\,\mathrm{K}$) CTTs are removed, i.e. the clouds that are easily categorised as liquid or ice by a CTT proxy such as $BT_{10.8}$ and for which a categorisation by the BTDs is therefore not necessary. Liquid clouds are

shown as round markers, while ice clouds are shown as crosses. The three subfigures in the left column of Fig. 11 vary only by their color code which encodes $\tau$, CTT and $R_{eff}$ respectively. They show that there is little overlap between the "typical" liquid and ice clouds (i.e. the clouds within the defined CTT and $R_{eff}$ boundaries). The only overlap is for very small $\tau$ ($\tau \lesssim 1$), since the BTDs approach the same values for all clouds, determined by atmospheric properties, as $\tau \to 0$ (best seen in Fig. 10). This means that a phase classification for "typical" liquid and ice cloud cases is possible in BTD(8.7-10.8) – BTD(10.8-12.0)

– space for $\tau \gtrsim 1$ when atmospheric parameters are known (see Sect. 6.3 for a short discussion on the generalisation to other atmospheric conditions).

    However, Fig. 10 and the left column in Fig. 11 also show that liquid and ice BTD values are closest for clouds with similar CTTs. To further explore this issue and to test the limitations of a phase classification using the BTDs, the right column of Fig. 11 shows BTD values also for clouds outside the "typical" cloud boundaries. The three subfigures show the whole range

of computed cloud scenarios (see table 1), including also exceptionally cold liquid clouds and exceptionally warm ice clouds. Only the "easy" to distinguish cases ($\tau \geq 10$ and either CTT$< 233\,\mathrm{K}$ or CTT $> 273\,\mathrm{K}$) are removed as before. The figures show that the overlap between liquid and ice clouds is significantly larger compared to the "typical" cloud cases (left column of Fig. 11). The clouds in the overlap region are mainly liquid and ice clouds which have similar CTTs in the midlevel temperature range, i.e. rather cold liquid clouds (CTT $\lesssim 260\,\mathrm{K}$) and rather warm ice clouds (CTT $\gtrsim 250\,\mathrm{K}$). We discussed in the last section

(Sect. 6.1) that the CTT/CTH is the most important contributor to the differences between liquid and ice clouds for both BTDs. It is therefore not surprising that phase discrimination for clouds with similar CTT/CTH is difficult even when knowledge of both BTDs is combined. Note also that additional information from $BT_{10.8}$, which is often used as a proxy for CTT, does not help much in distinguishing between phases in these cases of midlevel CTTs. For the phase classification of these midlevel clouds, the $R_{eff}$ also plays a role: For $R_{eff}$ values that are rather large for the respective phase, the overlap occurs for all $\tau$

values; for $R_{eff}$ values that are rather small for the respective phase, the overlap occurs only for very small or very large $\tau$ values.

    To summarize the most important results:

  – The combined use of BTD(8.7-10.8) and BTD(10.8-12.0) is better suited for phase discrimination than the two BTDs individually.





– The combined use of BTD(8.7-10.8) and BTD(10.8-12.0) can discriminate cloud phase for liquid and ice clouds in their "typical" CTT regimes as long as $\tau$ is not too small ($\tau \gtrsim 1$) and when atmospheric parameters are known.

   – Clouds in the midlevel CTT regime are challenging: If liquid clouds are particularly cold or ice clouds particularly warm, they can often not be distinguished by the two BTDs. This is especially true for clouds with large $R_{\text{eff}}$ for the respective phase.

## 6.3   Generalisation of findings

The RT results shown in this study are valid for the setup specified in table 1, for instance a cloud geometric thickness of 1 km, satellite zenith angle (SATZ) of 0° (nadir view) and the US standard atmosphere. In this section we briefly discuss how to generalize our results to different setups.

For constant $\tau$, a larger cloud geometric thickness means that radiation originates from deeper within the cloud (in terms
of geometric depth, implying a larger temperature difference). This depth can differ for different wavelengths, leading to a dependence of BTDs on geometric thickness (Piontek et al., 2021a). We conducted a sensitivity analysis for varying cloud geometric thickness between 1 km and 4 km. Results of this analysis are shown in Fig. E1 in the appendix. We find that the sensitivity to geometric thickness is comparably small ($\lesssim 0.5$ K), except for BTD(10.8-12.0) for cases of liquid clouds with very small $R_{\text{eff}}$, where the sensitivity to geometric thickness can exceed 1 K. However, this does not significantly influence
the regions in the space spanned by the two BTDs which are associated with the different phases or show an overlap between phases. Hence, it does not significantly influence a potential phase retrieval.

Changing SATZ to higher values has two effects on the BTD curves discussed in this study: Firstly, looking at clouds from an angle increases the path length through the cloud and therefore effectively increases the cloud optical thickness. Hence, all BTD curves as functions of $\tau$ are shifted to the left towards lower $\tau$ for both BTDs so that the BTD arcs as functions of $\tau$ are
narrowed. The second effect of an increased SATZ is a longer path through the atmosphere (above cloud). Since BTD(8.7-10.8) is sensitive to molecular absorption, it is therefore also sensitive to SATZ. For each CTH, this SATZ dependence introduces a shift towards lower BTD(8.7-10.8) values. The SATZ should therefore be taken into account when applying a phase retrieval.

For the same reasons, the atmospheric composition, especially the water vapor amount, can play a role especially for BTD(8.7-10.8): More water vapor induces lower BTD(8.7-10.8) values. This shift is greater the more water vapor is above
cloud top, and therefore greater the lower the CTH.

## 7   Conclusions

The aim of this study is to characterize and physically understand the relation of two IR window BTDs that are typically used for satellite retrievals of the thermodynamic cloud phase. As an example, we select (BTD(8.7-10.8) and BTD(10.8-12.0) of the SEVIRI imager, but the main findings can be generalised to other imagers with similar thermal channels. This knowledge
helps to design optimal cloud phase retrievals and to understand their potential and limitations. To this end, we present RT calculations that analyse the sensitivities of the two BTDs to cloud phase and all radiatively important cloud parameters related



to phase, namely $\tau$, $R_{eff}$, ice crystal habit and CTT/ CTH. Previous studies of BTDs have tended to focus on only a small number of cloud parameters, and an overview of the relative importance of all cloud parameters and their interdependencies is still missing. We perform a sensitivity analysis of the BTDs, which to our knowledge has never been done for all cloud
parameters combined. This provides an overview over the effects of all cloud parameters and shows which parameters are responsible for the observed phase dependence of the BTDs, which is often used for phase retrievals (Ackerman et al., 1990; Strabala et al., 1994; Finkensieper et al., 2016; Key and Intrieri, 2000; Baum et al., 2000, 2012; Mayer et al., 2024).

To understand the behaviour of the BTDs, we examine the effects of the nonlinear relationship between radiances and BTs through Planck's radiation law on the BTDs. This nonlinearity induces positive BTD values and a dependence on the CTT
in a simple RT model, even when cloud optical properties (transmittance and emissivity) are the same at all wavelengths. This effect is co-responsible for the arc shape of the BTDs as functions of $\tau$ and their CTT dependence, in addition to effects due to spectral differences in cloud optical properties. These spectral differences in cloud optical properties can explain the (remaining) dependence of the BTDs on the different cloud parameters.

We find that the dependence on phase is more complex than is sometimes assumed: Although both BTDs are directly
sensitive to phase (holding every other cloud parameter constant), this sensitivity is mostly small compared to other cloud parameters, as $\tau$, CTT and $R_{eff}$. Instead, apart from $\tau$ for which the sensitivity is well known, the BTDs show the strongest sensitivity to CTT/ CTH. Since the CTT is associated with phase, this is the main factor leading to the observed phase dependence of the BTDs. The direct phase dependence merely adds to the CTT/CTH effect, increasing differences between ice and liquid (for BTD(8.7-10.8) only for small $\tau \lessapprox 10$).

The sensitivity analysis shows that it is straightforward to distinguish "typical" high ice clouds from low liquid clouds using the BTDs. However, it is challenging to distinguish a mid-level ice cloud from a mid-level liquid cloud - especially if the $R_{eff}$ is also similar. The combination of both BTDs increases phase information content and is therefore preferable in a retrieval.

This study focuses on liquid and ice clouds. We expect the BTD values of mixed phase clouds to lie between ice and liquid values, as they represent a transition between the two. Depending mainly on the CTT/ CTH and to a lesser extent the $R_{eff}$ of
mixed phase clouds, their BTD values are expected to be closer or further away from the liquid or ice BTD values. Therefore, if the CTT/ CTH and $R_{eff}$ values are similar between liquid/mixed or mixed/ice, we expect the regions of the different phases to overlap in the space spanned by BTD(8.7-10.8) and BTD(10.8-12.0). Overall, we expect the BTDs to be useful in retrieving mixed-phase clouds, especially for clouds with distinct temperature/altitude and particle size regimes. However, for clouds with different phases that overlap in these cloud parameters, the use of more satellite channels containing information for instance
on particle size or phase would be beneficial to increase the information content.

*Code and data availability.* The libRadtran software used for the radiative transfer simulations is available from http://www.libradtran.org (Mayer and Kylling, 2005; Emde et al., 2016).



## Appendix A: Mathematical framework of radiative transfer

We briefly describe the mathematical framework of radiative transfer and the connection of the radiative transfer equations to the cloud and atmosphere is described mathematically by the radiative transfer equation

$$\frac{dL}{ds} = -(\beta_{\text{abs}} + \beta_{\text{sca}})L + \frac{\beta_{\text{sca}}}{4\pi}\int\limits_{4\pi} p(\Omega, \Omega')L(\Omega')d\Omega'$$

$$+ \beta_{\text{abs}}B(T),$$

where $L$ is the radiance along path $s$, $B(T)$ is the Planck function at temperature $T$, $\Omega$ is the solid angle, $\beta_{\text{abs}}$ and $\beta_{\text{sca}}$ are absorption and scattering coefficients and $p$ is the phase function. The three quantities $\beta_{\text{abs}}$, $\beta_{\text{sca}}$ and $p$ depend on the microphysical parameters $R_{\text{eff}}$, habit and cloud phase. The asymmetry parameter $g$ can be computed from $p$ as the average cosine of the scattering angle $\theta$. The asymmetry parameter $g$ characterizes the anisotropy of the scattering process: If a particle scatters more in the forward direction ($\Omega = 0°$), $g$ is positive; $g$ is negative if the scattering is more in the backward direction ($\Omega = 180°$) (Bohren and Huffman, 2008).

The optical thickness is connected to $\beta_{\text{abs}}$ and $\beta_{\text{sca}}$ by the relation

$$\tau(s_0, s_1) := \int\limits_{s_0}^{s_1} (\beta_{\text{abs}}(s) + \beta_{\text{sca}}(s))ds.$$

Defining $L_{sca} := \frac{\beta_{\text{sca}}}{4\pi}\int_{4\pi} p(\Omega, \Omega')L(\Omega')d\Omega'$ - i.e. the radiance scattered from all other directions into the direction of interest $\Omega$ - the radiative transfer equation (RTE) has the formal solution

$$L(s_1) = L(s_0)e^{-\tau(s_0, s_1)}$$

$$+ \int\limits_{s_0}^{s_1} ds\left[L_{sca}(s) + \beta_{\text{abs}}(s)B(T(s))\right]e^{-\tau(s, s_1)}.$$

For radiation traveling from the surface at $s_0$ through a cloud to the TOA at $s_1$, we can split the integral into its contributions from inside the cloud and its contribution from the atmosphere

$$L(s_1) \approx L(s_0)e^{-\tau(s_0, s_1)}$$

$$+ \int\limits_{s_0}^{s_{CB}} ds\,\beta_{\text{abs}}B(T)e^{-\tau(s, s_1)}$$

$$+ \int\limits_{s_{CB}}^{s_{CT}} ds\left[L_{sca} + \beta_{\text{abs}}B(T)\right]e^{-\tau(s, s_1)}$$

$$+ \int\limits_{s_{CT}}^{s_1} ds\,\beta_{\text{abs}}B(T)e^{-\tau(s, s_1)},$$



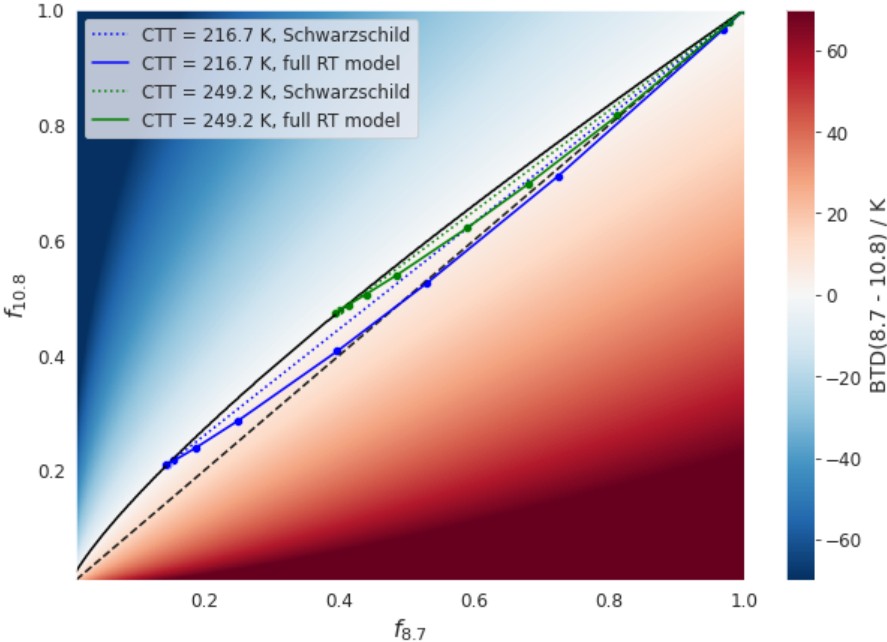

**Figure B1.** BTD(8.7-10.8) in the space spanned by the radiance fraction $f_{8.7}$ and $f_{10.8}$ (defined as the radiance at TOA scaled by the Planck radiance of the surface with temperature $T_s = 290\,\mathrm{K}$: $f_\lambda = R_{TOA,\lambda}/B_\lambda(T_s)$). The black solid line indicates BTD(8.7-10.8) = 0; the black dashed line indicates $f_{8.7} = f_{10.8}$. The blue and green lines show $f_{8.7}$ and $f_{10.8}$ values for varying $\tau$ at a given CTT: The dotted lines show $f_{8.7}$ and $f_{10.8}$ computed with the Schwarzschild equation (with $\tau_{8.7} = \tau_{10.8}$); solid lines show $f_{8.7}$ and $f_{10.8}$ values computed with the full RT model.

where $s_{CB}$ is at cloud bottom and $s_{CT}$ at cloud top. Since molecular scattering is negligible in the infrared window, we set $L_{sca} \approx 0$ in the two contributions from the atmosphere. The radiation at TOA $L(s_1)$ consists of (1) radiation emitted by the surface and atmosphere below the cloud (first and second term in the equation), (2) radiation emitted by the cloud or scattered by cloud particles in the direction of interest (third term in the equation) and (3) radiation emitted by the atmosphere above cloud top (fourth term in the equation; usually the smallest contribution compared to the other terms). Each contribution is weighted by the amount that is transmitted through the remaining material (atmosphere and cloud) along the ray.

## Appendix B: Disentangling the BTD Nonlinearity Shift from effects of wavelength dependent optical properties

An instructive way to look at the BTD Nonlinearity Shift and to disentangle it from effects of wavelength dependent optical properties is the following: To make the radiances at different wavelengths more comparable, we use the Planck radiance corresponding to the surface temperature $T_s$ as a reference. For typical atmospheric profiles (without temperature inversions), this Planck radiance $B_\lambda(T_s)$ is the maximal possible radiance in each wavelength, corresponding to $\tau \to 0$ (see Eq. 6). We





express the TOA radiance as fractions $f_\lambda$ of this maximal possible radiance, called *radiance fraction* in the following, i.e.

$$f_\lambda = \frac{R_{TOA,\lambda}}{B_\lambda(T_s)}, \quad \text{with } f_\lambda \in [0,1]. \tag{B1}$$

The BTDs can then be expressed as functions of the radiance fractions $f_\lambda$

$$\text{BTD}(\lambda_0 - \lambda_1) = T_{\lambda_0}(f_{\lambda_0} B_{\lambda_0}(T_s)) - T_{\lambda_1}(f_{\lambda_1} B_{\lambda_1}(T_s)). \tag{B2}$$

For the sake of brevity, in the following we only discuss BTD(8.7-10.8) as function of $f_{8.7}$ and $f_{10.8}$; BTD(10.8-12.0) has qualitatively the same properties and the same conclusions apply. Figure B1 shows BTD(8.7-10.8) in $f_{8.7}$-$f_{10.8}$-space for $T_s = 290\,\text{K}$. If $f_{8.7}$ is (much) larger than $f_{10.8}$, the BTD is positive and if $f_{8.7}$ is (much) smaller than $f_{10.8}$, the BTD is negative, as

expected. However, the BTD(8.7-10.8) $= 0$ line is not at $f_{8.7} = f_{10.8}$ (black dashed line in Fig. B1) as one might naively expect but has a convex shape in $f_{8.7}$-$f_{10.8}$-space (shown as black solid line), such that BTD(8.7-10.8) $= 0$ for $f_{8.7} < f_{10.8}$. Or to put it another way, if the radiance at TOA is the same fraction of its maximal possible radiance at both wavelengths, $f_{8.7} = f_{10.8}$, the BTD is positive. Note that this is a completely general statement, that does not depend on a RT model but simply shows what happens mathematically when the inverse Planck function, $T_\lambda$, is applied on fractions of Planck radiance, $f_\lambda B_\lambda(T_s)$, at

different wavelengths.

To understand the role of the BTD Nonlinearity Shift we add results of RT computations to Fig. B1 in the following steps: First, we study how radiances computed with the Schwarzschild equation look like in $f_{8.7}$-$f_{10.8}$-space. To see the pure BTD Nonlinearity Shift we again set the optical thickness constant at both wavelengths, $\tau_{8.7} = \tau_{10.8} = \tau$. Next, we explore the changes in the Schwarzschild radiance when $\tau$ differs at the two wavelengths, i.e. $\tau_{10.8} \neq \tau_{8.7}$. In this case, both the mathe-

matical BTD Nonlinearity Shift and the physical effect of spectrally dependent optical properties are present. Third, we study how the radiance computed with the full RT model looks like in $f_{8.7}$-$f_{10.8}$-space.

We start with the Schwarzschild radiance in $f_{8.7}$-$f_{10.8}$-space with constant optical thickness at both wavelengths, $\tau_{8.7} = \tau_{10.8} = \tau$. We compute the radiances $R_{\text{TOA},8.7}^{\text{S}}$ and $R_{\text{TOA},10.8}^{\text{S}}$ from the Schwarzschild equation as functions of $\tau$ for different values of CTT, as before (see Fig. 3(b)). These radiance results, expressed as radiance fractions $f_{8.7}$ and $f_{10.8}$, are shown in

Fig. B1 as dotted lines for two different CTTs. For $\tau = 0$, the TOA radiance is the radiance emitted by the surface and $f_{8.7} = f_{10.8} = 1$. As $\tau$ increases, $f_{8.7}$ and $f_{10.8}$ get smaller and BTD(8.7-10.8) $> 0$, since $f_{10.8}$ and $f_{8.7}$ show a linear relationship and the BTD(8.7-10.8) $= 0$ line is convex. For large $\tau$ ($\tau = 30$) the TOA radiance approaches the radiance emitted by a black body with a temperature equal to the CTT. Hence, the radiance fractions for large $\tau$ depend on the CTT and lie on the BTD(8.7-10.8) $= 0$ line (see Fig. B1). Overall, for increasing $\tau$ from 0 to 30, the Schwarzschild radiance fractions form a line from

$f_{8.7} = f_{10.8} = 1$ to the radiance fraction values corresponding to the CTT black body radiance. It follows from the convex shape of the BTD(8.7-10.8) $= 0$ line that lower CTTs lead to larger BTD(8.7-10.8) values (see Fig. B1). The fact that the Schwarzschild radiance fraction line deviates from the BTD(8.7-10.8) $= 0$ line such that BTD(8.7-10.8) $> 0$, depending on the CTT, is a representation of the BTD Nonlinearity Shift equivalent to Fig. 3(b). The property that $f_{10.8}$ is a linear function of $f_{8.7}$ can be shown from the Schwarzschild equation. Solving Eq. 6 for $e^{-\tau}$ for a given wavelength $\lambda_0$ and inserting it into the



Schwarzschild equation for a second wavelength $\lambda_1$ gives

$$R^{\mathrm{S}}_{\mathrm{TOA},\lambda_1} = k + m\, R^{\mathrm{S}}_{\mathrm{TOA},\lambda_0}, \tag{B3}$$

with

$$m = \frac{B_{\lambda_1}(T_s) - B_{\lambda_1}(\mathrm{CTT})}{B_{\lambda_0}(T_s) - B_{\lambda_0}(\mathrm{CTT})}, \tag{B4}$$

$$k = B_{\lambda_1}(T_s) - B_{\lambda_0}(T_s)\, m, \tag{B5}$$

i.e. a linear relationship between $R_{TOA,\lambda_1}$ and $R_{TOA,\lambda_0}$ and therefore also between $f_{\lambda_1}$ and $f_{\lambda_0}$.

So far we have set $\tau$ constant for all wavelengths in the Schwarzschild equation. To see what happens in the Schwarzschild model for different $\tau$ at different wavelengths, i.e. $\tau_{\lambda_0} \neq \tau_{\lambda_1}$, we add a small perturbation to $\tau_{\lambda_1}$,

$$\tau_{\lambda_1} = \tau_{\lambda_0} + \delta\tau. \tag{B6}$$

Since the Schwarzschild equation neglects scattering, $\tau_\lambda$ is determined by the absorption coefficient $\beta_{\mathrm{abs},\lambda}$ and the cloud water
path. For $\lambda_1 = 10.8\,\mu\mathrm{m}$ and $\lambda_0 = 8.7\,\mu\mathrm{m}$, the absorption coefficients $\beta_{\mathrm{abs},8.7} < \beta_{\mathrm{abs},10.8}$, meaning that if scattering is neglected $\tau_{8.7} < \tau_{10.8}$ and $\delta\tau > 0$ for this case. Solving Eq. 6 for a given $\lambda_0$ analogue to above for $e^{-\tau_{\lambda_0}}$ and inserting into Eq. 6 for $\lambda_1$ gives

$$R^{\mathrm{S}}_{\mathrm{TOA},\lambda_1} = k + m\, R^{\mathrm{S}}_{\mathrm{TOA},\lambda_0} - \delta\tau\, e^{-\tau_{\lambda_0}}\left(B_{\lambda_1}(T_s) - B_{\lambda_1}(\mathrm{CTT})\right), \tag{B7}$$

where we used $e^{-\delta\tau} \approx 1 - \delta\tau$. Hence, since $\delta\tau > 0$ for $\lambda_1 = 10.8\,\mu\mathrm{m}$ and $\lambda_0 = 8.7\,\mu\mathrm{m}$, $R^{\mathrm{S}}_{\mathrm{TOA},10.8}$ decreases when we add
a perturbation $\tau_{10.8} = \tau_{8.7} + \delta\tau$. This makes physical sense, since a larger $\tau_{10.8}$ compared to $\tau_{8.7}$ means that less radiance is transmitted through the cloud at 10.8 compared to $8.7\,\mu\mathrm{m}$. The amount by which $R^{\mathrm{S}}_{\mathrm{TOA},10.8}$ decreases is determined by the difference between surface and cloud top radiance, $B_{\lambda_1}(T_s) - B_{\lambda_1}(\mathrm{CTT})$, and the factor $\delta\tau\, e^{-\tau_{\lambda_0}}$. For $\tau_{\lambda_0} \to 0$, meaning that the cloud water path approaches zero, $\delta\tau \to 0$. For large $\tau_{\lambda_0}$, $e^{-\tau_{\lambda_0}} \to 0$. Hence, the last term in Eq. B7 vanishes for very small or large $\tau_{\lambda_0}$. For the $\tau_{\lambda_0}$ values in between, the perturbation $\delta\tau$ leads to a decrease of $R^{\mathrm{S}}_{\mathrm{TOA},10.8}$ and therefore of $f_{10.8}$. As a
result, the Schwarzschild radiance fraction line in $f_{8.7}$-$f_{10.8}$-space deviates from a linear to a concave line. This deviation is stronger for larger $\delta\tau$ (i.e. larger differences between $\tau_{10.8}$ and $\tau_{8.7}$), as well as for larger differences between the surface and the cloud top radiance, $B_{\lambda_1}(T_s) - B_{\lambda_1}(\mathrm{CTT})$.

As a last step of this analysis, we study the full RT model in $f_{8.7}$-$f_{10.8}$-space. Recall that in the full RT model, in general, $\tau_{8.7} \neq \tau_{10.8} \neq \tau$, where $\tau$ as usually refers to the optical thickness at 550 nm. Figure B1 shows the radiance fractions $f_{8.7}$ and
$f_{10.8}$ computed with the full RT model for an ice cloud for varying $\tau$ and two different CTTs in blue and green solid lines. Molecular absorption is switched off for these examples. Note that this is an equivalent representation of BTD(8.7-10.8) as the corresponding CTT curves in Fig. 8. For increasing $\tau$ from 0 to 30, the radiance fractions of the full RT model form curves from $f_{8.7} = f_{10.8} = 1$ to the radiance fraction values corresponding to the black body radiance of their CTT. These curves are concave, as expected from our theoretical considerations above (see Eq. B7). This concave shape, as explained above, can be



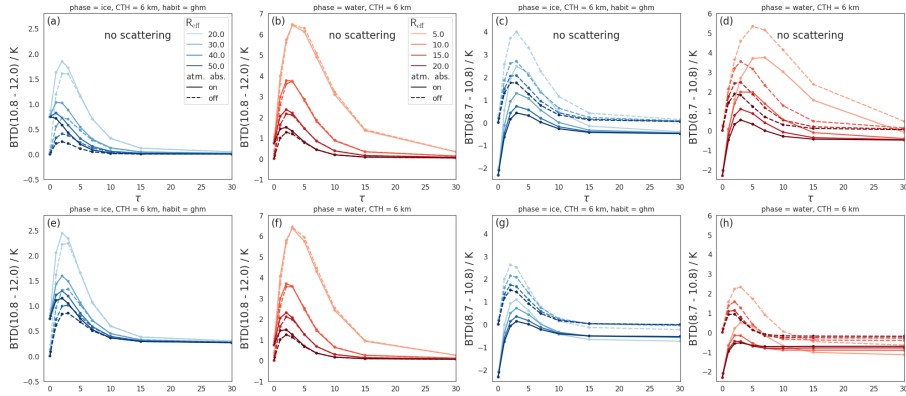

**Figure C1.** Effects of varying $R_{eff}$ on BTD(10.8-12.0) and BTD(8.7-10.8) as functions of $\tau$ for ice clouds (blue) and liquid clouds (orange/red) and scattering switched off (top row) and switched on (bottom row). Solid lines indicate a "normal" absorbing atmosphere, dashed lines indicate that molecular absorption is switched off.

attributed to differences in the absorption coefficients of the two wavelengths, $\beta_{abs,8.7} < \beta_{abs,10.8}$. The concave shape results in higher BTD values compared to the Schwarzschild BTD$^S$ values, where $\tau_{8.7} = \tau_{10.8} = \tau$ (compare BTD(8.7-10.8) along the solid and dotted lines in Fig. B1). The figure also shows that the deviation from the linear Schwarzschild radiance fraction lines is larger for lower CTTs - in accordance with our theoretical considerations (see Eq. B7).

This leads to the following interpretation of Fig. B1: The Schwarzschild radiance fraction lines in Fig. B1 (dotted lines)
represent the pure BTD Nonlinearity Shift, which induces positive BTD values even though $\tau$ is the same in all wavelengths. Adding spectral differences between the cloud optical properties "pushes" the radiance fraction lines into a concave shape and further increases BTD. Hence, the difference between the BTD(8.7-10.8) = 0 line and the Schwarzschild radiance fraction lines in Fig. B1 is due to the Nonlinearity of the transformation from radiances to BTs; the difference between the Schwarzschild radiance fraction lines and the full RT model (solid lines) in Fig. B1 is due to the spectral differences in cloud optical proper-
ties. Lower CTTs increase both the BTD Nonlinearity Shift and the effects of spectral differences between the cloud optical properties.

## Appendix C: Effects of $R_{eff}$ on BTDs - disentangling absorption and scattering effects

Figure C1 shows the sensitivity of both BTDs with $R_{eff}$ broken down into effects of absorption and scattering. The two rows show the same cloud scenarios, once with scattering switched off (top row) and once with scattering switched on (bottom row).
The figure shows that the effects of absorption lead to increasing values for smaller $R_{eff}$ for both BTDs (top row of Fig. C1).

For BTD(10.8-12.0), the effect of scattering is similar for varying $R_{eff}$ and comparatively small (increases (decreases) BTD(10.8-12.0) by $\lesssim 0.5\,\text{K}$ for ice (water) clouds; compare Fig. C1(a,b) with (e,f)). For BTD(8.7-10.8), scattering effects are stronger than for BTD(10.8-12.0) and depend on $R_{eff}$: Scattering leads to a stronger decrease of BTD(8.7-10.8) for smaller



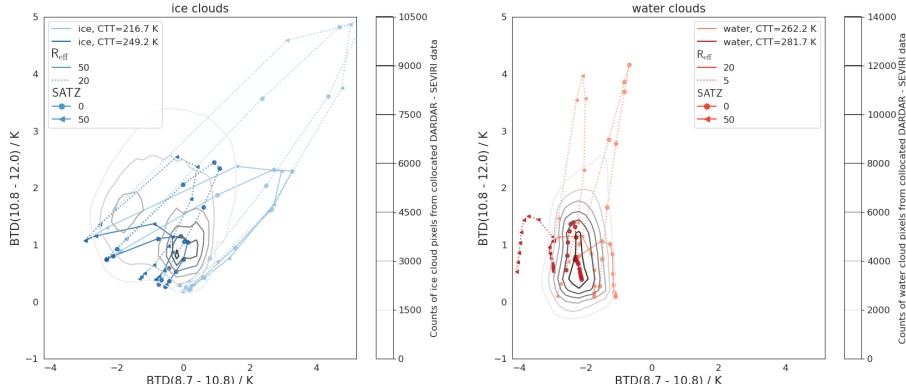

**Figure D1.** Comparison of RT results with measured SEVIRI data. The RT results are displayed as in Fig. 10, but for a fixed ice crystal habit (ghm) and two SATZ values (different markers). The corresponding counts of measured SEVIRI data is overlayed as contours in grey.

$R_{eff}$ (compare Fig. C1(c,d) with (g,h)). Since, however, the absorption effects are stronger, BTD(8.7-10.8) increases with
decreasing $R_{eff}$ (Fig. C1(g,h)).

## Appendix D:  Comparison to measured Satellite data

Figure D1 shows a comparison of the RT results with measured SEVIRI data. The SEVIRI data was collocated with the active satellite product DARDAR (Delanoë and Hogan, 2010) containing information on the cloud phase (for more details see Mayer et al. (2023)). The plot on the left shows ice clouds; the plot on the right water clouds. As in Sect. 6.1 and Sect. 6, the RT results
show boundary cases of "typical" cloud scenarios in blue and red, as indicated in the legend. In addition to SATZ=0°, we also show the RT results for SATZ=50°, in order to be able to compare the RT results to a large number of measurements with angles between these two cases. The measured SEVIRI data with the corresponding constraints (i.e. data of ice or water clouds within CTT and SATZ boundaries as for the RT calculations) are plotted on top of the RT results in grey. The figure shows that the RT results and measured SEVIRI data have a large overlap. Hence, the computed ranges of BTD values are realistic.

**Appendix E:  Effects of cloud geometric thickness on BTDs**

Figure E1 shows a sensitivity analysis for varying cloud geometric thickness between 1 km and 4 km. The sensitivity to geometric thickness is comparably small and mostly $\lesssim 0.5$ K. An exception are liquid clouds with very small $R_{eff}$ for BTD(10.8-12.0), where the sensitivity to geometric thickness can exceed 1 K.

For the case of liquid clouds with CTT = 281.7 K the CTH is at an altitude of 1 km in the US-standard atmospheric profile.
Geometric thicknesses > 1 km are therefore not possible in this case.

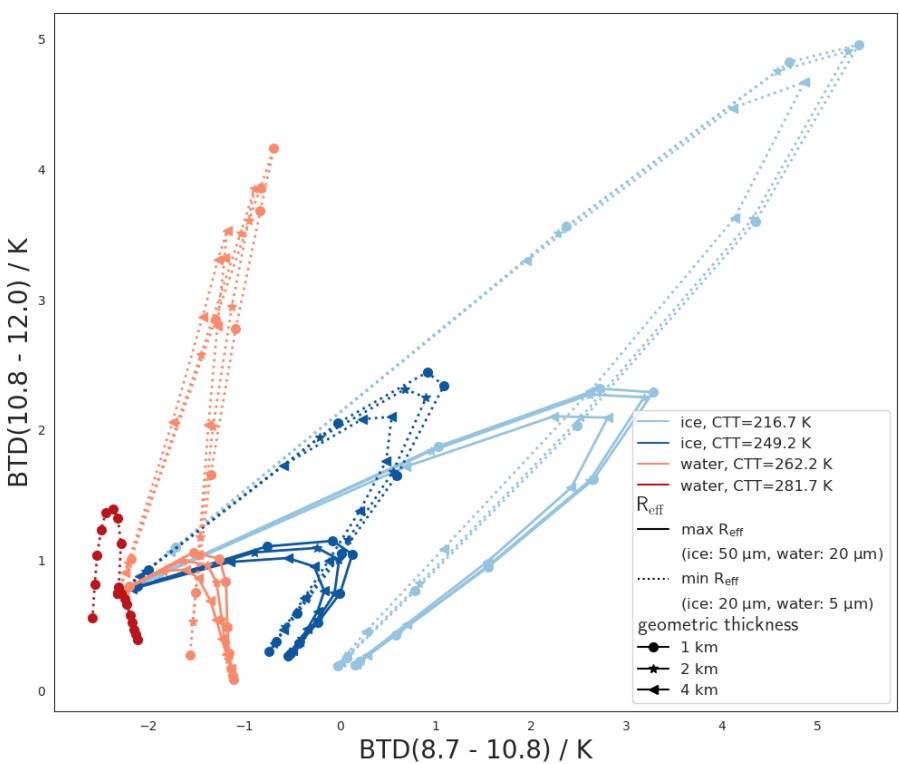

**Figure E1.** Same as Fig. 10, but for a fixed ice crystal habit (ghm) and varying geometric thickness of the cloud (in different markers)

*Author contributions.* All authors contributed to the project through discussions. JM carried out the simulations and the analysis of the data with valuable feedback from LB, BM and RM. LB and CV supervised the project and provided scientific feedback. JM took the lead in writing the manuscript. All authors provided feedback on the manuscript.

*Competing interests.* The authors declare that they have no conflict of interest.

*Acknowledgements.* We thank Klaus Gierens for constructive discussions and valuable feedback. This research was funded by the Deutsche Forschungsgemeinschaft (DFG, German Research Foundation)–TRR 301–Project-ID 428312742.



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
