# Peer review of "How well can brightness temperature differences of spaceborne imagers help to detect cloud phase? A sensitivity analysis regarding cloud phase and related cloud properties"

_EGUsphere, 2024_

## Referee Comment (RC1)

**Reviewer Report for egusphere-2024-540**

**Paper Summary**

This study in investigates cloud property sensitivities of two IR BTD cloud phase tests based on simulations of SEVIRI radiances at three IR window channels. The authors rigorously delineate the role of cloud phase, Reff, tau, and ice habit which are related to the spectral differences in bulk single scattering properties when those cloud parameters are changed. The study focuses on a single ocean scene with a fixed Ts and atmospheric profile. The authors also investigate the connected roles of CTT and the non-linear radiance-to-brightness-temperature conversion on the BTDs. Both nonlinearity effect and decrease in CTT introduced positive BTDs. The BTDs showed most sensitivity to CTT, followed by tau, the BTD non-linearity effects and spectral dependent single scattering properties of ice and liquid that change with particle size and ice habit assumptions. Sensitivity studies are also performed using the bounding properties (e.g., tau, CTT) of realistic mid-latitude cloud properties. The author's also compare their BTDs with SEVIRI observations to further justify the realism of the BT spectra. Overall, the study illustrates that BTD-based phase retrievals are complex, and the factors influencing the BTD are not only due to differences in cloud microphysical properties and optical depth.

**Review Summary**

This paper is well written and well thought out. In terms of justifications for this study (as written in the intro), the modeling methods, and the overall conclusions, I do not have many comments. The results of the study also seem very reasonable and the author's explain them in a lot of detail. In fact, I would even suggest trying to reduce some detail and possible redundancies in the results section. The results section was a bit hard to get through, mainly in differentiating sections 5 and 6. The comments below should clarify some of my confusions regarding the results section. Overall, this is a solid paper and I recommend it for publication after the relatively minor comments below are addressed by the authors.

**Major Comments:**

1. I believe the title of the paper could be modified slightly to better align with the methodology of the paper and be more specific and accurate. I suggest replacing "*information content*" with "*sensitivity analysis*" in the title. "*Information Content*" may give the reader the impression that a mathematical information content analysis will be performed with a metric such as the commonly used Shannon Entropy (Shannon & Weaver [1949]). Also, since the study is only focused on a particular ocean case with a standard US atmosphere, it would be beneficial to write something like "*Over Mid-Latitude Oceans*" into the title. Additionally, "*with respect to cloud phase*" seems slightly misleading, since a number of cloud properties are examined, not only phase. The sensitivity study seems more comprehensive than just the role of cloud phase in the BTDs. I suggest changing the words "*cloud phase*" to something like "*cloud phase and other properties*". I think it would improve the title's accuracy and scope.

2. Sections 5 and 6 appear to perform very similar analyses and they could be consolidated into a single section in order to avoid confusion for the reader. For example, figure 8 shows BTDs for a range of ice cloud CTTs, which are nearly identical to the realistic

range of CTTs for ice clouds used in Section 6. It seems that Section 6 is providing (1) an emphasis on the BTDs expected for the realistic boundaries of observed cloud properties and (2) comparisons of the BTDs together over the range of realistic cloud scenes (e.g. Figure 10). If that is the case, please make that clear as you transition from Figure 6-8 to Figure 9 and beyond.

3. The title of section 6.3 in the main paper is somewhat confusing, because a "*generalization*" of the findings would presumably include more discussion than just for cloud geometric thickness and viewing geometry effects on BTDs, in my view. A simple way to address this issue, is to change the title of the section to reflect the specific content of this subsection (i.e., cloud thickness and viewing geometry). Furthermore, I am not sure how much value the cloud geometric thickness discussion adds to the paper overall, as this should be including more vertical variations in temperature and humidity within the cloud layer, which does not appear to be in the scope of the paper. I recommend removing the discussion on geometric thickness in section 6.3, and moving the satellite viewing geometry discussion to the appendix.

4. On line 548, the author's write "*Overall, we expect the BTDs to be useful in retrieving mixed-phase cloud*". It is well known that identifying mixed phase clouds with passive remote sensing is extremely challenging, if not impossible in some cases. The author's state on line 544: "*We expect the BTD values of mixed phase clouds to lie between ice and liquid values, as they represent a transition between the two.*" The mixed phase cloud BTDs being in between the ice and liquid BTD solution spaces introduces ambiguity when trying to differentiate between liquid, ice and mixed phase clouds using the BTD approach. Furthermore, mixed phase clouds can exist at the temperatures in which both ice and supercooled liquid clouds can exist, which further complicates the use of BTDs for mixed phase cloud classification (based on the relationship between CTT and BTDs). I suggest that the author's further justify in the conclusion their claim that the SEVIRI BTDs can be used for identification of mixed phase clouds or modifying the statements in the conclusion that speculate on mixed phase cloud classification success.

5. Throughout the paper, the sensitivity of BTDs to CTTs is emphasized, and rightly so. However, the sensitivity to CTT is more accurately described as a sensitivity to the thermal contrast between the cloud top and surface if I am not mistaken. In polar regions, for example, the thermal contrast tends to be very low and CTTs can be low as well, and this BTD approach becomes less useful. I suggest that the authors make sure to emphasize the cloud-surface-temperature contrast impact (in addition to CTT) clearly in the abstract and conclusions of the paper.

6. I recommend adding a clear statement in the conclusion that emphasizes that the study focuses on a single ocean scene with a fixed atmosphere, and the results shown could change depending on the scene (e.g., Tropical vs. Subarctic). This can be followed by a brief discussion how variations in surface types (surface emissivity), surface temperature, and atmospheric temperature and humidity (discussed already in the results section) may impact the conclusions of the paper. The results of the paper already provide information

for this additional discussion, and the inclusion of this would make the paper more complete, in my view. A few sentences would be sufficient.

**Minor Comments:**
1. Figure 3: Can the authors place a horizontal black line to the left of the "tau=0.5" in the upper left legend of panel a? That would make a clear connection between the tau value and the black curve. It would also be beneficial for the reader if the authors added the CTT for the lower radiance curve B(CTT) to the figure, as this can provide context for the lower panels. Also adding the BT values to the figure (for the SEVIRI channels) with the corresponding colors would help readers to better understand the arguments being made.

2. Table 1: Can you include the total column water vapor amount for the US standard atmosphere? It would be useful for context when "switching on/off" molecular absorption.

3. Line 247: "*This means that for a given τ in the figures the water content is held constant for the scenario with and without scattering*". I'm not sure why the water content statement is there, because changing the water content would change the physical cloud. Optical thickness is the quantity being held constant here for each scattering scenario.

4. Figure 5: It would be beneficial to include something like "*w/scattering*" for the lower panels, just for the reader's benefit. Or the panels can be arranged to have no scattering on one column and w/scattering for the other column.

5. Line 435: "*about −2K in the figure*". What figure are the authors referring to?

6. Figure 10: It could be beneficial to include example BTD threshold lines or joint BTD solution spaces for liquid and ice phase here (perhaps from one of the BTD algorithm references that were cited in the intro). This gives the reader an intuition for how the BTDs are typically used for phase classification, and how confounding factors like CTT may lead to shifts into or out of ice and liquid BTD solution spaces. It also provides clarity on why mixed phase cases are so difficult with this method.

7. Did you use SEVIRI spectral response functions in your radiative transfer simulations? If so, please specify. If not, please mention that in the radiative transfer methods section.

8. Section 5.1: Can the authors make a comment on why you did not use different habits in the scattering sensitivity analyses? The overall conclusions may not change, but a sentence mentioning how the results would change if the ice habit was changed would be helpful.

9. Line 87: *"These findings help to better understand and improve the working principles of phase retrieval algorithms".* To be more accurate, could the authors write as "*brightness temperature difference phase retrieval algorithms*"?

10. Figure 6: Above the panels, could the authors use CTT instead of CTH? Using CTT provides the readers for more relevant context, as the results can be compared to other figures with the same CTT as used in this Figure. I recommend doing this for other figures that may show CTH instead of CTT.

11. Line 438 "*Overall, the phase information contained in BTD(8.7-10.8) comes mainly from its sensitivity to CTT for clouds with $\tau \lesssim 10$ and from its sensitivity to CTH".* Since the CTH and CTT are directly correlated in the authors experimental setup mentioning CTH here seems redundant.

12. I suggest writing units into relevant figure legends where they are missing, or specify in the captions (e.g., Reff legend values in Figure 6).

13. Line 546: When the author's write "*Therefore, if the CTT/ CTH and Reff values are similar between liquid/mixed or mixed/ice…".* I am not sure what the author's mean by liquid/mixed and ice/mixed here. It doesn't seem to be defined anywhere in the paper. Please clarify.

---

## Author Comment (AC1)

Dear Reviewer,

thank you for taking the time to review our manuscript. Your feedback is greatly appreciated and was helpful in improving the quality of this research. We value your constructive criticism and thoughtful comments, which have helped to identify areas that require further clarification and refinement.

We carefully considered your suggestions and incorporated them into the revised manuscript to address the issues raised, as specified below (referee comments in black; our answers in blue).

**Comments of Reviewer #2**

I think the previous two reviewers had very nice summary and evaluation of the manuscript. I agree with a lot of their comments. Overall I think this is a very rigorous study of the IR based retrieval of cloud properties and it is worthy of publication. On the other hand I think the manuscript needs some modification and revision before it can be finally accepted for publication.

First of all the manuscript reads more like a shortened Ph.D./MS. thesis than a scientific paper. It is way too long and hard to get insightful/meaningful results without having to go through many unnecessary parts. For example, the section 2 "physics background" should be significantly shortened if not removed completely as it is about something that is from textbook and well known. Similarly, I'm not sure if the Appendix A is needed either. In any case, I would suggest finding all possible ways to make the paper succinct.

Thank you for this feedback, it is very valuable to know. We have made several changes to the manuscript to shorten it and make the messages and results clearer.

We have significantly shortened section 2 "Physical Background" and moved the explanations of the single scattering properties to the appendix. Although much of the physical background is well known, we believe it is valuable to keep the parts which are most essential for understanding the content of the paper for readers less familiar with the subject.

We agree that the content of Appendix A (Mathematical Framework of Radiative Transfer) is well known and not essential for understanding our study. We have removed it as suggested.

We have also shortened the results sections (sections 5 and 6) to make them more concise. The largest changes have been made to Section 6.1 to avoid redundancy with Section 5 and to make its messages clearer. The shortened version of Section 6.1 is shown in the response to Major Comment 2 from Reviewer #1.

The discussion of "disentangling the roles of cloud absorption and scattering" is interesting. However, it is based on the assumption that scattering is completely ignored in the IR cloud remote sensing, which is rarely the case in recent studies. As mentioned by the first reviewer, a popular method is the so-called beta ratio method, which was first proposed Parol et al., 1991 and then improved and used in many follow up studies (e.g., Pavolonis 2010, Heidinger et al. 2010; Heidinger et al., 2015;). The beta ratio method is based on the so-called similarity principle that takes into account the strong forward scattering of cloud particles in radiative transfer. I think it is important to point this out (with proper reference) and discuss it against Eq. (6).

Sorry for the confusion: We did not want to give the impression that scattering is ignored in IR cloud remote sensing - IR retrievals are often based on RT calculations (e.g. look-up tables), which usually also simulate scattering. Our intention in studying the different roles of absorption and scattering is to better understand the underlying physics of how these processes contribute to the BTD values. We believe that this aspect is not often studied in detail and we want to contribute to a better explanation of why the BTDs show certain behavior. We do not assume that Eq. (6), which only considers absorption, is the basis for recent IR retrievals. Eq. (6) is only intended to demonstrate the BTD nonlinearity effect in its simplest form.

To make it very clear that scattering is usually not ignored in IR remote sensing, we rewrote two sentences in the introduction:

"Radiative transfer through clouds and the atmosphere is complex, with many parameters that can in principle influence satellite observations. Although radiative transfer models are capable to correctly account for all of these quantities, the relative importance of these parameters is often not fully understood."

"In addition, the origin of the dependence of BTDs on cloud thermodynamic phase, as observed in satellite measurements and radiative transfer results, is not fully understood. Although phase retrievals are usually based on accurate radiative transfer calculations that take into account all radiative effects, it is argued that variations in the refractive indices of ice and water across the infrared window cause the BTDs to be sensitive to cloud phase (Finkensieper et al. 2016, Key et al. 2000, Baum et al. 2000, Baum et al. 2012)."

Furthermore, we agree that the beta ratio method is very interesting and has many strengths. We have added a note in the introduction of our manuscript that beta ratios are another popular approach to cloud phase retrievals. However, we believe that a detailed discussion of beta ratios is beyond the scope of this study, which is explicitly focused on BTDs and their sensitivities. We added the following to our manuscript:

"Finally, we note that besides BTDs, there are other popular methods for retrieving cloud phase and other cloud properties, such as β ratios (Parol et al., 1991; Pavolonis, 2010; Heidinger et al., 2015). While this study is specifically aimed at BTDs, understanding the effects of different cloud properties on the radiative transfer through clouds is also useful to better understand the physics underlying β ratio retrievals."

While the manuscript covers most of the important aspects of the IR based retrieval of cloud properties, an important missing is the impacts of cloud microphysics. A good paper on this topic is Zhang et al (2010). I think it would make the study stronger if you can add some study/discussion to about this effect, perhaps along with the discussion on the effects of cloud geometrical thickness.

If we understand correctly, you are suggesting a discussion on the impact of vertical inhomogeneity of microphysical parameters on BTDs since this is the topic of the Zhang et al. (2010) paper and the impact of cloud microphysics (ice habit, Reff) is already included in our manuscript. We agree that this is a very interesting topic and that an additional discussion of the effects of vertical inhomogeneity would make the study more complete. We have therefore added a sensitivity analysis

of the BTDs to the vertical inhomogeneity of $R_{eff}$ (see figure below) and added it to the discussion of the effects of cloud geometric thickness in the Appendix, as suggested:

[revised manuscript text omitted]

Heidinger, A. K., M. J. Pavolonis, R. E. Holz, B. A. Baum, and S. Berthier (2010), Using CALIPSO to explore the sensitivity to cirrus height in the infrared observations from NPOESS/VIIRS and GOES-R/ABI, *J. Geophys. Res.*, 115, D00H20, doi:10.1029/2009JD012152.

Heidinger, A.K.; Li, Y.; Baum, B.A.; Holz, R.E.; Platnick, S.; Yang, P. Retrieval of Cirrus Cloud Optical Depth under Day and Night Conditions from MODIS Collection 6 Cloud Property Data. Remote Sens. **2015**, 7, 7257-7271. https://doi.org/10.3390/rs70607257

Zhang, Z., S. Platnick, P. Yang, A. K. Heidinger, and J. M. Comstock (2010), Effects of ice particle size vertical inhomogeneity on the passive remote sensing of ice clouds, *J. Geophys. Res.*, 115, D17203, doi:10.1029/2010JD013835

---

## Author Comment (AC2)

Dear Reviewer,

thank you for taking the time to review our manuscript. Your feedback is greatly appreciated and was helpful in improving the quality of this research. We value your constructive criticism and thoughtful comments, which have helped to identify areas that require further clarification and refinement.

We carefully considered your suggestions and incorporated them into the revised manuscript to address the issues raised, as specified below (referee comments in black; our answers in blue).

**Comments of Reviewer #1**

**Paper Summary**

This study in investigates cloud property sensitivities of two IR BTD cloud phase tests based on simulations of SEVIRI radiances at three IR window channels. The authors rigorously delineate the role of cloud phase, Reff, tau, and ice habit which are related to the spectral differences in bulk single scattering properties when those cloud parameters are changed. The study focuses on a single ocean scene with a fixed Ts and atmospheric profile. The authors also investigate the connected roles of CTT and the non-linear radiance-to-brightness-temperature conversion on the BTDs. Both nonlinearity effect and decrease in CTT introduced positive BTDs. The BTDs showed most sensitivity to CTT, followed by tau, the BTD non-linearity effects and spectral dependent single scattering properties of ice and liquid that change with particle size and ice habit assumptions. Sensitivity studies are also performed using the bounding properties (e.g., tau, CTT) of realistic mid-latitude cloud properties. The author's also compare their BTDs with SEVIRI observations to further justify the realism of the BT spectra. Overall, the study illustrates that BTD-based phase retrievals are complex, and the factors influencing the BTD are not only due to differences in cloud microphysical properties and optical depth.

**Review Summary**

This paper is well written and well thought out. In terms of justifications for this study (as written in the intro), the modeling methods, and the overall conclusions, I do not have many comments. The results of the study also seem very reasonable and the author's explain them in a lot of detail. In fact, I would even suggest trying to reduce some detail and possible redundancies in the results section. The results section was a bit hard to get through, mainly in differentiating sections 5 and 6. The comments below should clarify some of my confusions regarding the results section. Overall, this is a solid paper and I recommend it for publication after the relatively minor comments below are addressed by the authors.

Thank you for the positive review of our manuscript. Regarding the aspects of reducing some details, we have made several changes to the manuscript to shorten it and to make the results section in particular more concise. We have also rewritten large parts of section 6.1 to avoid redundancy with section 5 and to make its messages clearer (see response to Major Comment 2 below and response to the report of reviewer #2).

**Major Comments:**

1. I believe the title of the paper could be modified slightly to better align with the methodology of the paper and be more specific and accurate. I suggest replacing "information content" with "sensitivity analysis" in the title. "Information Content" may give the reader the impression that a mathematical information content analysis will be performed with a metric such as the

commonly used Shannon Entropy (Shannon & Weaver [1949]). Also, since the study is only focused on a particular ocean case with a standard US atmosphere, it would be beneficial to write something like "Over Mid Latitude Oceans" into the title. Additionally, "with respect to cloud phase" seems slightly misleading, since a number of cloud properties are examined, not only phase. The sensitivity study seems more comprehensive than just the role of cloud phase in the BTDs. I suggest changing the words "cloud phase" to something like "cloud phase and other properties". I think it would improve the title's accuracy and scope.

We agree with several of the points made, such as using "sensitivity analysis" instead of "information content" and emphasizing that other cloud parameters besides phase are studied.

The reviewer is right that we focus on a particular setup of atmosphere and an ocean surface. Note, however, that the chosen atmospheric/surface/viewing geometry setup does not represent realistic conditions (e.g. a satellite zenith angle of zero implies a tropical atmosphere instead of the US-standard atmosphere used in our study). Instead this setup is chosen to be relatively simple to be able to focus on the influence of cloud parameters. We therefore think that adding "Over Mid Latitude Oceans" to the title would be misleading, as it implies a realistic scenario. Furthermore, the aim of this study is to characterize and physically understand the relation of cloud parameters to the BTDs. The main findings of this study, including the physical understanding of the effects of cloud properties on BTDs and their relative importance, are valid for any atmospheric or surface condition. For this reason, we have chosen not to include any specific information about atmospheric or surface conditions in the title.

Keeping these aspects and the reviewer's suggestions in mind, we changed the title to: "How well can brightness temperature differences of spaceborne imagers help to detect cloud phase? A sensitivity analysis regarding cloud phase and related cloud properties"

2. Sections 5 and 6 appear to perform very similar analyses and they could be consolidated into a single section in order to avoid confusion for the reader. For example, figure 8 shows BTDs for a range of ice cloud CTTs, which are nearly identical to the realistic range of CTTs for ice clouds used in Section 6. It seems that Section 6 is providing (1) an emphasis on the BTDs expected for the realistic boundaries of observed cloud properties and (2) comparisons of the BTDs together over the range of realistic cloud scenes (e.g. Figure 10). If that is the case, please make that clear as you transition from Figure 6-8 to Figure 9 and beyond.

Thank you for this feedback, it is very valuable to know that there is some confusion regarding the distinction of Section 5 and 6. In Section 5 we analyse the effects of various individual cloud properties on the two individual BTDs, by varying only one cloud property at a time, in order to improve the physical understanding of these effects. In Section 6 we assess the combined effects of all cloud parameters and look at the BTDs from the perspective of a phase retrieval: we analyse for which cloud scenarios we can distinguish between liquid clouds and ice clouds, and when they overlap. To avoid confusion for the reader we rewrote the introductory text of Section 6 as follows:

"In the last section we analysed the effects of cloud properties on the BTDs individually, by varying only one cloud property at a time (besides $\tau$ ). In this section we combine the phase related cloud parameters $\tau$, $R_{eff}$ , ice habit, CTT and thermodynamic phase for a sensitivity analysis of the BTDs. From this analysis we determine typical BTD ranges for ice and liquid clouds and understand which cloud parameters are responsible for the phase information contained in the BTDs. We analyse for which cloud scenarios we can distinguish between liquid clouds and ice clouds, and when they overlap, allowing us to derive implications for phase retrievals. First, in Sect. 6.1, we perform sensitivity analyses for each BTD individually. Next, in Sect. 6.2, we study the sensitivities and phase information content of the two BTDs combined."

We have also shortened Section 6.1 to avoid redundancy with Section 5 and to make its messages clearer. The main results part of Section 6.1 now reads as follows (note that former Fig. 9 is now Fig. 10):

"In Figure 10 BTD(10.8-12.0) shows the highest sensitivity to τ , CTT and Reff. BTD(8.7-10.8) shows the highest sensitivity to τ, CTT and molecular absorption (closely linked to CTH). In comparison to τ and CTT/CTH the sensitivity to Reff is lower for BTD(8.7-10.8) and mainly relevant for small CTT. For both BTDs, the direct sensitivity to cloud phase, i.e. holding all other cloud parameters constant, plays mostly only a minor role: For BTD(10.8-12.0) the direct phase dependence is of the order of 0.5–1.5 K; for BTD(8.7-10.8) the direct influence of phase is only significant for small τ values ($\lesssim$ 10) and then of the order of 1–2 K (see Sect. 5.3).
For a phase retrieval we need to know for which cloud properties liquid and ice clouds overlap and where they separate for both BTDs. The largest BTD(10.8-12.0) values in the "typical" cloud scenarios (about 2.5 to 5 K in Fig. 10) are only observed for optically thin and cold ice clouds with small Reff. Thus BTD(10.8-12.0) is useful to detect cirrus clouds, especially if they have small Reff (like contrails), and classify them as ice in a phase retrieval. However, our calculations show that certain liquid cloud scenarios with exceptionally low Reff and cold CTTs can also induce remarkably high BTD(10.8-12.0). This can lead to misclassification of these liquid clouds as ice. However, most liquid clouds have lower BTD(10.8-12.0), below about 2.5 K in Fig. 10. Since such low BTD(10.8-12.0) may also indicate ice clouds with "warm" CTTs and/or large Reff , or ice clouds with τ close to zero, a phase classification based on BTD(10.8-12.0) alone is challenging. The lowest BTD(10.8-12.0) values (about 0 to 1 K in Fig. 10) indicate optically thick clouds, but do otherwise not contain much phase information.
As for BTD(10.8-12.0), large BTD(8.7-10.8) (around 1 to 5.5 K in Fig. 10) can indicate ice phase, since only ice clouds with low τ of about 1 < τ < 7 reach these values. Low BTD(8.7-10.8) (lower than about −0.5 in Fig. 10) can arise from very thin ice clouds (as BTD(8.7-10.8) decreases to about -2 K as τ goes to zero) or optically thick clouds. For optically thick clouds, BTD(8.7-10.8) decreases with higher CTT (due to lower CTHs and stronger molecular absorption) and smaller Reff - both characteristics typical of liquid clouds. As a general guideline for optically thick clouds, lower BTD(8.7-10.8) indicate a higher probability of a liquid cloud. Overall, the phase information contained in BTD(8.7-10.8) originates mainly from its sensitivity to CTT for clouds with τ $\lesssim$ 10, while for optically thick clouds it stems mainly from its sensitivity to molecular absorption (closely linked to CTH) and (to a lesser extent) Reff. Only in cases of optically thin clouds (τ $\lesssim$ 10) is the phase information of BTD(8.7-10.8) additionally due to the direct phase influence on the (different) absorption properties of liquid and ice particles."

3. The title of section 6.3 in the main paper is somewhat confusing, because a "generalization" of the findings would presumably include more discussion than just for cloud geometric thickness and viewing geometry effects on BTDs, in my view. A simple way to address this issue, is to change the title of the section to reflect the specific content of this subsection (i.e., cloud thickness and viewing geometry). Furthermore, I am not sure how much value the cloud geometric thickness discussion adds to the paper overall, as this should be including more vertical variations in temperature and humidity within the cloud layer, which does not appear to be in the scope of the paper. I recommend removing the discussion on geometric thickness in section 6.3, and moving the satellite viewing geometry discussion to the appendix.

Thank you for these suggestions. We have made several changes regarding the content of this section:
First, we moved the (shortened) discussion about the satellite viewing geometry and some additional points about effects of different atmospheric and surface setups to the conclusion (see response to Major Comment 6). We believe that it is important to emphasize that the

results may change for different atmospheric/surface/viewing geometry setups, as you rightly also recommend in Major Comment 6.
Second, because the discussion of atmospheric/surface/viewing geometry setups has been moved, Section 6.3 is now a discussion only about the effects of additional cloud parameters, namely cloud geometric thickness and vertical Reff inhomogeneity (as suggested by the second reviewer). We have kept this discussion very brief by moving most of it to the Appendix and keeping only the main results relevant to the sensitivity of the BTDs in Section 6.3. We believe that including an estimate of the effects of these additional cloud parameters makes our study more complete, since one of its goals is to provide an overview of the relative importance of different cloud parameters for the BTDs. Since the focus of the study is on cloud properties, we believe that a discussion of geometric thickness falls within the scope of the study, while varying atmospheric parameters (temperature, humidity, ...) are out of scope. To reflect the updated content of Section 6.3, we have changed the title to "Sensitivity to additional cloud parameters": Effects of geometric thickness and vertical $R_{eff}$ inhomogeneity". Section 6.3 now reads as the following:

"Cloud properties that have not been discussed so far are cloud geometric thickness and vertical inhomogeneities of microphysical parameters. Both can have an impact on BTDs (Piontek et al., 2021a; Zhang et al., 2010). To estimate how large these effects are, we performed a sensitivity analysis for varying cloud geometric thickness and for vertical inhomogeneities of $R_{eff}$. Results of this analysis are shown in Fig. E1 and Fig. E2 in the appendix. We find that the sensitivity to both geometric thickness and vertical $R_{eff}$ inhomogeneity is small compared to other cloud parameters ($\lesssim$ 0.5 K in most cases). This sensitivity does not significantly affect the regions in the space spanned by the two BTDs which are associated with the different phases and therefore has a comparatively small effect on a potential phase retrieval."

4. On line 548, the author's write "Overall, we expect the BTDs to be useful in retrieving mixed-phase cloud". It is well known that identifying mixed phase clouds with passive remote sensing is extremely challenging, if not impossible in some cases. The author's state on line 544: "We expect the BTD values of mixed phase clouds to lie between ice and liquid values, as they represent a transition between the two." The mixed phase cloud BTDs being in between the ice and liquid BTD solution spaces introduces ambiguity when trying to differentiate between liquid, ice and mixed phase clouds using the BTD approach. Furthermore, mixed phase clouds can exist at the temperatures in which both ice and supercooled liquid clouds can exist, which further complicates the use of BTDs for mixed phase cloud classification (based on the relationship between CTT and BTDs). I suggest that the author's further justify in the conclusion their claim that the SEVIRI BTDs can be used for identification of mixed phase clouds or modifying the statements in the conclusion that speculate on mixed phase cloud classification success.

We completely agree with the referee's comment that identifying mixed phase clouds with passive remote sensing is challenging and that ice, mixed-phase and liquid clouds can overlap in the space spanned by the BTDs, introducing ambiguity. With our statement in line 548 we wanted to say that the BTDs can be one helpful factor in the detection of mixed-phase clouds; we did not want to imply that they can detect mixed-phase clouds without ambiguity. As we write in our manuscript in line 546: "…if the CTT/ CTH and Reff values are similar between liquid/mixed or mixed/ice, we expect the regions of the different phases to overlap in the space spanned by BTD(8.7-10.8) and BTD(10.8-12.0)."

To avoid confusion, we have reformulated the paragraph to make it clearer:
"This study focuses on liquid and ice clouds. We expect the BTD values of mixed-phase clouds

to lie between ice and liquid values, as they represent a transition between the two. Depending mainly on the CTT/ CTH and to a lesser extent the Reff of mixed-phase clouds, their BTD values are expected to be closer or further away from the liquid or ice BTD values. In that sense, we expect that BTDs can make a useful contribution to the retrieval of mixed-phase clouds and their composition. However, as the CTT/ CTH and Reff values overlap between liquid, mixed-phase and/or ice clouds, we expect the regions of the different phases in the space spanned by BTD(8.7-10.8) and BTD(10.8-12.0) to also overlap, introducing ambiguity. The use of additional satellite channels containing, for instance, particle size or phase information is necessary to increase the phase information content for a retrieval."

5. Throughout the paper, the sensitivity of BTDs to CTTs is emphasized, and rightly so. However, the sensitivity to CTT is more accurately described as a sensitivity to the thermal contrast between the cloud top and surface if I am not mistaken. In polar regions, for example, the thermal contrast tends to be very low and CTTs can be low as well, and this BTD approach becomes less useful. I suggest that the authors make sure to emphasize the cloud-surface-temperature contrast impact (in addition to CTT) clearly in the abstract and conclusions of the paper.

We agree that the observed CTT dependence more generally is mainly a sensitivity to the surface-cloud temperature contrast, and have followed the suggestion to emphasize this more clearly in the manuscript. We have therefore made various small additions at several points in the manuscript, as detailed below.

In the abstract:
"Instead, the primary link between phase and the BTDs lies in their sensitivity to CTT (or more generally the surface-cloud temperature contrast $\Delta T$), which is associated with phase."

In Section 4 (on the BTD nonlinearity shift):
"… Thus, even if $\tau_\lambda$ is the same for all three wavelengths, $\tau_\lambda = \tau$, the nonlinearity of the inverse Planck function induces positive BTDS values and a dependence on the CTT. More generally, this dependence is mainly a sensitivity to the thermal contrast $\Delta T := T_s -$ CTT; however, for a fixed $T_s$, as shown in the examples here, it reduces to a dependence on CTT."

We had already mentioned in subsection 5.6 (on CTT dependence) that the observed CTT dependence is mainly a dependence on the surface-cloud temperature contrast. We have added more emphasis to this fact in the last paragraph of Section 5.6:
"Hence, the effects of spectral differences in optical properties on BTDS are amplified by larger $\Delta T$, i.e. differences between $T_s$ and the CTT. This is the main reason (besides the BTD Nonlinearity Shift) for the CTT dependence of the BTDs. Colder CTTs (or rather larger $\Delta T$) thus increase both the BTD Nonlinearity Shift and the effects of spectral differences in optical properties"
We also emphasized this fact in the bullet point summary at the end of Section 5.6.

We added the following sentence to the conclusions:
"Note that more generally, this CTT dependence of the BTDs is more accurately described as a dependence on the surface-cloud temperature contrast $\Delta T$, which reduces to a CTT dependence in our case with a fixed surface temperature."

6. I recommend adding a clear statement in the conclusion that emphasizes that the study focuses on a single ocean scene with a fixed atmosphere, and the results shown could change depending on the scene (e.g., Tropical vs. Subarctic). This can be followed by a brief discussion how variations in surface types (surface emissivity), surface temperature, and atmospheric

temperature and humidity (discussed already in the results section) may impact the conclusions of the paper. The results of the paper already provide information for this additional discussion, and the inclusion of this would make the paper more complete, in my view. A few sentences would be sufficient.

We agree that this additional discussion is valuable. We have added a brief discussion in the conclusions:

"This study was conducted for a simple fixed setup of the atmosphere, surface and satellite viewing geometry in order to focus on the effects of cloud properties. If this setup is changed, we expect the cloud effects on the BTDs discussed in this paper to be superimposed by additional effects: For example, changes in water vapor content or satellite zenith angle shift BTD(8.7-10.8) due to its sensitivity to water vapor absorption. This shift is larger the more water vapor is above the cloud top and therefore depends on the CTH and the vertical atmospheric profile. A different type of surface with spectral differences in surface emissivity (as for instance a desert surface) shifts the values of both BTDs for optically thin clouds. For potential phase retrievals, these effects should ideally be taken into account."

**Minor Comments:**

1. Figure 3: Can the authors place a horizontal black line to the left of the "tau=0.5" in the upper left legend of panel a? That would make a clear connection between the tau value and the black curve. It would also be beneficial for the reader if the authors added the CTT for the lower radiance curve B(CTT) to the figure, as this can provide context for the lower panels. Also adding the BT values to the figure (for the SEVIRI channels) with the corresponding colors would help readers to better understand the arguments being made.

We followed the suggestions and made the following changes to the figure: We added the values of the CTT and $T_S$ to the lower and upper radiance curves B(CTT) and B($T_S$) and added "tau = 0.5" to the black curve. Further, we added labels to the dashed curves with the BT values in the legend (see figure below).

[Figure]

2. Table 1: Can you include the total column water vapor amount for the US standard atmosphere? It would be useful for context when "switching on/off" molecular absorption.

Thank you for the suggestion, we agree that adding the total column water vapor is useful for context. We added the total column water vapor amount of 14.3 kg/m2 for the US-standard atmosphere in table 1.

3. Line 247: "This means that for a given τ in the figures the water content is held constant for the scenario with and without scattering". I'm not sure why the water content statement is there, because changing the water content would change the physical cloud. Optical thickness is the quantity being held constant here for each scattering scenario.

Sorry for the confusion. What we want to do here is compare two physically identical clouds (same water content, Reff, ...), one with scattering switched on and one with scattering switched off. We know that switching off scattering in a cloud changes its optical thickness, since only absorption now contributes to the extinction of radiation: for two microphysically identical clouds (i.e. same IWC and Reff), the optical thickness is smaller for the cloud in which scattering is switched off. To compare the two clouds, we therefore use the "original" optical thickness τ (with scattering) for both clouds on the x-axis of the figure, i.e. the same water content at any given τ in the figure.

We explain this in the manuscript in the sentences before the line in question. However, we feel that the sentence in question was confusing and removed it. We rewrote the explanation as follows:

"Switching off scattering in a cloud changes the optical thickness of that cloud, since only absorption now contributes to the extinction of radiation. However, to be able to compare scenarios with and without scattering for fixed cloud microphysics (same water content, $R_{eff}$, …), the τ parameter used for this figure is still the "original" optical thickness (with absorption and scattering)."

4. Figure 5: It would be beneficial to include something like "w/scattering" for the lower panels, just for the reader's benefit. Or the panels can be arranged to have no scattering on one column and w/scattering for the other column.

We understand that it would be beneficial for the reader to include "with scattering" in panels (c) and (d), in order to more quickly grasp the difference between the different rows. However, we believe that this could lead to confusion as to whether or not scattering is included in the following figures, as we do not include "with scattering" there either. We would therefore like to keep the figure as it is. We believe that the caption of Fig. 5 (now Fig. 6) is unambiguous and should resolve any confusion the reader may have: "Brightness temperature differences BTD(10.8-12.0) and BTD(8.7-10.8) as functions of τ for cloud particle scattering (a,b) switched off and (c,d) switched on for liquid and ice clouds".

5. Line 435: "about −2K in the figure". What figure are the authors referring to?

We were referring to Figure 9, sorry for the confusion. In the revised version of Section 6.1, the sentence and reference in question is removed (see above in the response to Major Comment 2).

6. Figure 10: It could be beneficial to include example BTD threshold lines or joint BTD solution spaces for liquid and ice phase here (perhaps from one of the BTD algorithm references that were cited in the intro). This gives the reader an intuition for how the BTDs are typically used for phase classification, and how confounding factors like CTT may lead to shifts into or out of ice and liquid BTD solution spaces. It also provides clarity on why mixed phase cases are so difficult with this method.

In principle, we like the idea. However, it is difficult to find suitable threshold lines for such a comparison: First, different methods use different thresholds. Second, many of the traditional BTD methods have been developed for MODIS, which may be somewhat misleading since, although the MODIS channels used are similar to the corresponding SEVIRI channels, their spectral response functions have some differences (Ackermann et al., 1990; Strabala et al., 1994; Baum et al., 2000). Third, and most importantly, most phase methods use not only BTDs but also additional other measures, so example thresholds for BTDs alone may be misleading (Strabala et al., 1994; Baum et al., 2012; Hünerbein et al., 2022; Benas et al., 2023; Mayer et al. 2024). To make this third point very clear, we added a comment in the conclusion of the manuscript:

"Although modern phase retrievals often rely not only on BTDs but also on other satellite measurements (Baum et al., 2012; Hünerbein et al., 2022; Benas et al., 2023; Mayer et al., 2024), it is important to understand the BTD characteristics and capabilities. This knowledge helps to design optimal cloud phase retrievals and to understand their potential and limitations."

7. Did you use SEVIRI spectral response functions in your radiative transfer simulations? If so, please specify. If not, please mention that in the radiative transfer methods section.

We have used parameterized SEVIRI spectral response functions, thank you for noticing that we did not specify this in the manuscript. We added it to Section 3 (Radiative transfer calculations):

"Simulations of TOA radiances for the SEVIRI IR window channels are made using the one-dimensional radiative transfer solver DISORT (Discrete Ordinate Radiative Transfer) 2.0 by Stamnes et al. (2000) and Buras et al. (2011) with parameterized SEVIRI channel response functions as described by Gasteiger et al. (2014)."

8. Section 5.1: Can the authors make a comment on why you did not use different habits in the scattering sensitivity analyses? The overall conclusions may not change, but a sentence mentioning how the results would change if the ice habit was changed would be helpful.

We agree that this additional information could be interesting for the reader and want to also consider the other cloud properties. We added a brief explanation that, as our calculations show, changing cloud properties (habit, CTT and Reff) would not impact the main messages of the paper:

"Using different CTT or Reff values in the calculations (for both the liquid and the ice cloud) mainly changes the magnitude of the negative peaks but does not change the qualitative results shown in Fig. 4. (now Fig. 5) Similarly, changing the ice crystal habit does not change the qualitative results and has only a small effect on the values shown."

9. Line 87: "These findings help to better understand and improve the working principles of phase retrieval algorithms". To be more accurate, could the authors write as "brightness temperature difference phase retrieval algorithms"?

We changed the sentence to "…working principles of phase retrieval algorithms that use BTDs…"

10. Figure 6: Above the panels, could the authors use CTT instead of CTH? Using CTT provides the readers for more relevant context, as the results can be compared to other figures with the same CTT as used in this Figure. I recommend doing this for other figures that may show CTH

instead of CTT.

*We followed the suggestion and used CTT instead of CTH in the figure panels everywhere, such that the figures are easier to compare to each other.*

11. Line 438 "Overall, the phase information contained in BTD(8.7-10.8) comes mainly from its sensitivity to CTT for clouds with $\tau \lesssim 10$ and from its sensitivity to CTH". Since the CTH and CTT are directly correlated in the authors experimental setup mentioning CTH here seems redundant.

*Here, we wanted to distinguish between the effects of the CTT (or the surface-cloud temperature contrast) and the effects of molecular absorption, which are directly connected to the CTH and are one of the most important effects for optically thick clouds for BTD(8.7-10.8). However, we understand that the wording was confusing and rewrote the sentence to (see also response to Major Comment 2):*

*"Overall, the phase information contained in BTD(8.7-10.8) comes mainly from its sensitivity to CTT for clouds with $\tau \lesssim 10$, while for optically thick clouds it comes mainly from its sensitivity to molecular absorption (closely linked to CTH) and (to a lesser extent) Reff."*

12. I suggest writing units into relevant figure legends where they are missing, or specify in the captions (e.g., Reff legend values in Figure 6).

*Thank you for noticing this mistake. We added relevant units in the legends where they were missing.*

13. Line 546: When the author's write "Therefore, if the CTT/ CTH and Reff values are similar between liquid/mixed or mixed/ice…". I am not sure what the author's mean by liquid/mixed and ice/mixed here. It doesn't seem to be defined anywhere in the paper. Please clarify.

*We wanted to describe situations where liquid CTTs and mixed-phase CTTs are similar, or where mixed-phase CTTs and ice CTTs are similar. We understand that the wording was confusing and we rewrote the sentence as follows:*

*"Therefore, if the CTT/ CTH and $R_{eff}$ values are similar between liquid, mixed-phase and/or ice clouds…"*

**References**

Ackerman, S. A., Smith, W. L., Revercomb, H. E., and Spinhirne, J. D.: The 27-28 October 1986 FIRE IFO Cirrus Case Study: Spectral Properties of Cirrus Clouds in the 8-12 μm Window, Monthly Weather Review, 118, 2377–2388, https://doi.org/10.1175/1520-0493(1990)118<2377:TOFICC>2.0.CO;2, 1990.

Baum, B. A., Menzel, W. P., Frey, R. A., Tobin, D. C., Holz, R. E., Ackerman, S. A., Heidinger, A. K., and Yang, P.: MODIS Cloud-Top Property Refinements for Collection 6, Journal of Applied Meteorology and Climatology, 51, 1145–1163, https://doi.org/10.1175/JAMC-D-11-0203.1, 2012.

Benas, N., Solodovnik, I., Stengel, M., Hüser, I., Karlsson, K.-G., Håkansson, N., Johansson, E., Eliasson, S., Schröder, M., Hollmann, R., and Meirink, J. F.: CLAAS-3: the third edition of the CM SAF cloud data record based on SEVIRI observations, Earth System Science Data, 15, 5153–5170, https://doi.org/10.5194/essd-15-5153-2023, 2023

Buras, R., Dowling, T., and Emde, C.: New secondary-scattering correction in DISORT with increased efficiency for forward scattering, Journal of Quantitative Spectroscopy and Radiative Transfer, 112, 2028–2034, https://doi.org/10.1016/j.jqsrt.2011.03.019, 2011.

Gasteiger, J., Emde, C., Mayer, B., Buras, R., Buehler, S., and Lemke, O.: Representative wavelengths absorption parameterization applied to satellite channels and spectral bands, Journal of Quantitative Spectroscopy and Radiative Transfer, 148, 99–115, https://doi.org/10.1016/j.jqsrt.2014.06.024, 2014.

Hünerbein, A., Bley, S., Horn, S., Deneke, H., and Walther, A.: Cloud mask algorithm from the EarthCARE multi-spectral imager: the M-CM products, https://doi.org/10.5194/egusphere-2022-1240, 2022.

Mayer, J., Bugliaro, L., Mayer, B., Piontek, D., and Voigt, C.: Bayesian Cloud Top Phase Determination for Meteosat Second Generation, EGUsphere, 2024, 1–32, https://doi.org/10.5194/egusphere-2023-2345, 2024.

Piontek, D., Bugliaro, L., Kar, J., Schumann, U., Marenco, F., Plu, M., and Voigt, C.: The New Volcanic Ash Satellite Retrieval VACOS Using MSG/SEVIRI and Artificial Neural Networks: 2. Validation, Remote Sensing, 13, 3128, https://doi.org/10.3390/rs13163128, 2021a.

Stamnes, K., Tsay, S.-C., Wiscombe, W., and Laszlo, I.: DISORT, a General-Purpose Fortran Program for Discrete-Ordinate-Method Radiative Transfer in Scattering and Emitting Layered Media: Documentation of Methodology, Tech. rep., Dept. of Physics and Engineering Physics, Stevens Institute of Technology, Hoboken, NJ 07030, 2000.

Strabala, K. I., Ackerman, S. A., and Menzel, W. P.: Cloud Properties inferred from 8-12-μm Data, 33, 212–229, https://doi.org/10.1175/1520-0450(1994)033<0212:CPIFD>2.0.CO;2, 1994.

---

## Author Comment (AC3)

**Response to community comment of Andrew Heidinger**

This paper is a very thorough sensitivity analysis of SEVIRI IR brightness temperature differences. It will serve as a reference for future studies. Excellent text and high quality figures. I urge its publication and my comments should be considered optional.

Thank you for your positive feedback on our study and for taking the time to write this comment. Your insights are greatly appreciated and we value your thoughtful comments and questions.

Line 30. Many algorithms use these brightness temperature differences along with other measurements (brightness temperature of other channels, reflectances …) to retrieve phase simultaneously with other cloud properties (temperature, reff, tau). Are you saying this is necessary?

Good point, our wording was too strong. We rewrote the sentence to
"Additionally, determining cloud phase is often a prerequisite in the remote sensing retrieval of cloud properties, including $\tau$ , Reff and water path (Marchant et al., 2016)."

Figure 2. SEVIRI offers channels at 9.7 and 13.3um but they occur in absorption bands. These seem to offer different single scattering properties that add to the phase story. Do you see any benefit of their inclusion or is the atmospheric absorption too much in your opinion?

As you correctly point out, the SEVIRI 9.7um channel occurs in an ozone absorption band. Ozone variations, e.g. due to its annual cycle, the ozone hole or planetary waves, can be significant and lead to large errors in cloud remote sensing (Ewald et al., 2013). Temperature variations in the stratosphere also add to the complexity. Overall, we believe that these factors make it difficult to use the 9.7um channel for cloud remote sensing.

The 13.4um channel is easier to use in this sense than 9.7, since $CO_2$ is well mixed in the troposphere and its annual cycle is less pronounced (Ewald et al., 2013). As the 13.4um channel is well suited to retrieve cloud top height (e.g. using $CO_2$ slicing), it could be helpful to determine a CTT and to get an estimate of the amount of absorbing atmosphere above the cloud (which is important for the interpretation of the 8.7um channel). We therefore believe that the inclusion of the 13.4um channel could be beneficial.

Line 40.0 You mention Parol (1991). One of the main tools of his analysis was to convert BTDs in beta ratios which are directly linked to the single scattering properties and remove many of the temperature, optical depth dependences. Do you have a position on the use of beta ratios in cloud phase determination?

We believe that beta ratios are a very powerful tool that goes beyond BTDs as they can be directly related to microphysical quantities (phase, habit, Reff). Since more information is used to compute beta ratios (RT to compute clear sky radiances, cloud temperature...), we expect that they will be able to discriminate cloud composition better than just using BTDs without any additional information.
However, we have not worked with beta ratios and have not investigated the sensitivities and capabilities/limitations of beta ratios. Nevertheless, we believe that some of our findings could also be applied to beta ratios, in particular the physical understanding of the effects of different cloud

properties on the radiative transfer through the cloud.

Line 355, I am confused by the two statements "the single scattering properties are not CTT dependent" and "cloud emissivity is not CTT dependent (as it is a function of the absorption coefficient)". Is not the absorption coefficient a single scatter property? Emissivity is solely a function of reff and tau unless you mean the "effective emissivity" observed through brightness temperatures.

Sorry for the confusion, our wording was a bit misleading. We wanted to say that the single scattering properties do not depend on CTT and that the effects of the CTT on BTDs can therefore not be traced back directly to spectral differences of the single scattering properties. We changed the sentence in the manuscript to the following:

"Since the single scattering properties are not CTT dependent (see Sect. 2), this CTT effect on the BTDs is also not (directly) due to spectral differences in the single scattering properties - in contrast to the effects of the other cloud parameters discussed above."

Line 490. One of the big issues in satellite imager cloud phase determination is the discrepancy in cloud phase from IR and VIS/NIR approaches for mid-level clouds. Is this something you see in your analysis?

We have not looked more closely at VIS/NIR retrievals and so cannot say much about them. However, one finding of our study is that the primary link between phase and BTDs is their sensitivity to CTT, which is associated with phase. As a consequence, it is easy to distinguish high ice clouds from low liquid clouds, but difficult to distinguish mid-level ice clouds from mid-level liquid clouds. Therefore, mid-level clouds are one of the most challenging cloud types for phase discrimination using BTDs.

Line 550: Is this study only to show these dependencies or is targeted for improving or understanding a current or future cloud phase product?

This study originated from calculations made for a new phase retrieval (Mayer et al., 2024). We wanted to better understand the role of the BTDs used in the retrieval and to optimize the information content gained from them. We hope that other existing or future phase retrieval methods will also benefit from this analysis.

Overall, excellent. It might add to the story to show a compelling SEVIRI scene or two.

Thank you very much. We like the suggestion to add a SEVIRI scene to give an example of how the BTD channel combinations look like. We have added an RGB composite and the two BTDs of the same scene to the introduction.

**References**

Ewald, F., Bugliaro, L., Mannstein, H., and Mayer, B.: An improved cirrus detection algorithm

MeCiDA2 for SEVIRI and its evaluation with MODIS, Atmospheric Measurement Techniques, 6, 309–322, https://doi.org/10.5194/amt-6-309-2013, 2013.

Marchant, B., Platnick, S., Meyer, K., Arnold, G. T., and Riedi, J.: MODIS Collection 6 shortwave-derived cloud phase classification algorithm and comparisons with CALIOP, Atmospheric Measurement Techniques, 9, 1587–1599, https://doi.org/10.5194/amt-9-1587-2016, 2016.

Mayer, J., Bugliaro, L., Mayer, B., Piontek, D., and Voigt, C.: Bayesian Cloud Top Phase Determination for Meteosat Second Generation, EGUsphere, 2024, 1–32, https://doi.org/10.5194/egusphere-2023-2345, 2024.